# SEIZING SERENDIPITY: EXPLOITING THE VALUE OF PAST SUCCESS IN OFF-POLICY ACTOR-CRITIC

## ABSTRACT

Learning high-quality $Q$-value functions plays a key role in the success of many modern off-policy deep reinforcement learning (RL) algorithms. Previous works focus on addressing the value overestimation issue, an outcome of adopting function approximators and off-policy learning. Deviating from the common viewpoint, we observe that $Q$-values are indeed underestimated in the latter stage of the RL training process, primarily related to the use of inferior actions from the current policy in Bellman updates as compared to the more optimal action samples in the replay buffer. We hypothesize that this long-neglected phenomenon potentially hinders policy learning and reduces sample efficiency. Our insight to address this issue is to incorporate sufficient exploitation of past successes while maintaining exploration optimism. We propose the Blended Exploitation and Exploration (BEE) operator, a simple yet effective approach that updates $Q$-value using both historical best-performing actions and the current policy. The instantiations of our method in both model-free and model-based settings outperform state-of-the-art methods in various continuous control tasks and achieve strong performance in failure-prone scenarios and real-world robot tasks[1].

## 1 INTRODUCTION

Reinforcement learning (RL) has achieved impressive progress in solving many complex decision-making problems in recent years (Mnih et al., 2015; Silver et al., 2016; Hutter et al., 2016; Ouyang et al., 2022). Many of these advances are obtained by off-policy deep RL algorithms, where the ability to leverage off-policy samples to learn high-quality value functions underpins their effectiveness. Value overestimation (Fujimoto et al., 2018; Moskovitz et al., 2021) has long been recognized as an important issue in off-policy RL, which is primarily associated with the function approximation errors (Fujimoto et al., 2018) and the side-effect of off-policy learning (Auer et al., 2008; Jin et al., 2018; Azar et al., 2017), and can potentially lead to suboptimal policies and sample inefficiency. To tackle this issue, techniques for alleviating the overestimation errors of value functions have been ubiquitously adopted in modern off-policy RL algorithms (Haarnoja et al., 2018a; Laskin et al., 2020; Han & Sung, 2021; Moskovitz et al., 2021).

Intriguingly, we find in this paper that in most online off-policy actor-critic methods, the commonly known value overestimation issue might disappear and be replaced by value underestimation when the agent gradually starts to solve the task[2]. A concrete example is illustrated in Figure 1a: in a failure-prone quadruped robot locomotion task DKittyWalkRandomDynamics (Ahn et al., 2020), SAC could underestimate historical successful experiences in the latter part of the training process. A possible explanation is that, the $Q$-value update could be negatively impacted by the actions $a'$ (*i.e.*, target-update actions) from the current suboptimal policy as compared to using samples from the scarce successful experiences in the replay buffer when computing the target $Q$-value $Q(s', a')$. If such circumstances occur, the RL agent would take a substantially longer time to re-encounter these serendipities for training with decreased sample efficiency.

---

[1]Please refer to https://beeauthors.github.io for experiment videos and benchmark results.
[2]For comprehensive investigations of the underestimation issue in the under-exploitation stage, please see Appendix E.

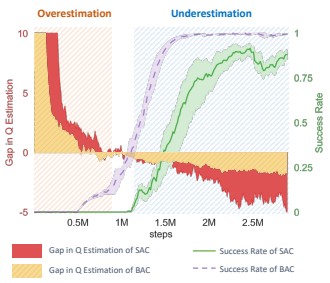 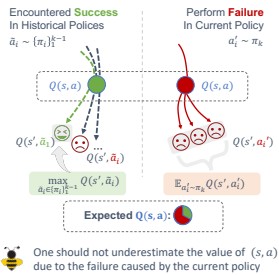

(a) Visualization on the underestimation issue.   (b) Illustrative figure on target-update actions.

Figure 1: Motivating examples. **(a)** In the DKittyWalkRandomDynamics task, when current policy generated action is inferior to the best action in the replay buffer, which usually occurs in the later stage of training (referred to as the under-exploitation stage), SAC is more prone to underestimation pitfalls than BAC. The gap in $Q$ estimation is evaluated by comparing the SAC $Q$-values and the Monte-Carlo $Q$ estimates using the trajectories in the replay buffer. **(b)** We expect the $Q$ value of $(s, a)$ that ever observed with the successful successor $(s', a')$ to be high. But the Bellman evaluation operator, whose target-update actions $a'$ are only sampled from the current policy, tends to underestimate it.

The above observation highlights the existence of an *under-exploitation* stage after the initial *under-exploration* stage (Aytar et al., 2018; Ecoffet et al., 2019; 2021) in many robotic tasks, where the $Q$-value can be underestimated due to the insufficient exploitation on the high-quality samples in the replay buffer (illustrated in Figure 1b). Thus allowing the RL agent to swiftly seize the serendipities, i.e., luckily successful experiences, can be a natural cure to resolve the underestimation issue without introducing additional overestimation, while also providing potentially improved sample efficiency.

At the heart of this paper, we connect this intuition with Bellman operators: the Bellman Exploitation operator enables effective exploitation of high-quality historical samples while the Bellman Exploration operator targets maintaining exploration optimism. A simple but effective mechanism, the Blended Exploration and Exploitation (BEE) operator, is then proposed to combine the merits of both sides. BEE operator can provide superior $Q$-value estimation, especially in addressing the "under-exploitation" issue. Moreover, it can be flexibly integrated into existing off-policy actor-critic frameworks, leading to the instantiations of two practical algorithms: BAC (BEE Actor-Critic) for model-free settings and MB-BAC (Model-based BAC) for model-based settings.

Both BAC and MB-BAC outperform other state-of-the-art methods on various MuJoCo, DMControl and Meta-World tasks by a large margin. On many failure-prone tasks such as DogRun and HumanoidStandup, BAC achieves over 2x the evaluation scores of the strongest baseline. Crucially, we conduct real-world experiments on four competitive quadruped robot locomotion tasks, and BAC achieves strong performance owing to the capability of addressing the under-exploitation issue. Furthermore, in our experiments, we observe unanimously improved performance when applying the BEE operator to different backbone algorithms, highlighting its flexibility and generic nature.

## 2 RELATED WORKS

Off-policy actor-critic methods leverage a replay buffer to update the $Q$-function and policy (Casas, 2017; Mnih et al., 2016), providing higher sample efficiency than on-policy RL methods. The prior works commonly rely on the standard policy gradient formulation (Peters & Schaal, 2008) for policy improvement. Various attempts have been devoted to modifying the policy evaluation procedure, primarily pursuing a high-quality value function to tackle the exploration or exploitation issue — central concerns in online RL (Burda et al., 2019; Ecoffet et al., 2019).

Despite the ongoing interest in exploration and exploitation, most previous works devote to exploration design following the optimism principle in the face of uncertainty (Auer et al., 2008; Fruit et al., 2018; Szita & Lőrincz, 2008), but view exploitation as merely maximizing $Q$-function. The Bellman evaluation operator, $\mathcal{T}Q(s, a) = r(s, a) + \gamma \mathbb{E}_{s' \sim P} \mathbb{E}_{a' \sim \pi} Q(s', a')$, underpins the critic learning. Existing efforts can be summarized into modifying this operator $\mathcal{T}$ in three main ways: 1) perturbing action $a'$ with techniques such as $\epsilon$-greedy, target policy smoothing (Fujimoto et al., 2018), and pink noise (Eberhard et al., 2023); 2) augmenting reward $r$ to foster exploration Ostrovski et al. (2017); Burda et al. (2019); Badia et al. (2020); Zhang et al. (2021b); 3) directly adjusting $Q$ values such

as max-entropy RL methods (Zhang et al., 2021a; Hazan et al., 2019; Lee et al., 2019; Islam et al., 2019; Haarnoja et al., 2018a; Han & Sung, 2021) that infuse the operator with an entropy term, and optimistic exploration methods that learn Upper Confidence Bound (UCB) (Ishfaq et al., 2021; Auer, 2002; Nikolov et al., 2019) of ensemble $Q$-value networks (Bai et al., 2021; Ciosek et al., 2019; Moskovitz et al., 2021). In essence, value overestimation might be associated with optimistic exploration (Jin et al., 2018; Laskin et al., 2020; Moskovitz et al., 2021), alongside factors such as off-policy learning and high-dimensional, nonlinear function approximation. Hence, attempts to correct for overestimation, *e.g.*, taking the minimum of two separate critics, have been widely adopted in the above exploration-driven methods (Fujimoto et al., 2018; Haarnoja et al., 2018a; Han & Sung, 2021; Sun et al., 2022). Yet directly applying such a minimum may cause underestimation (Hasselt, 2010). To mitigate it, prior methods (Ciosek et al., 2019; Moskovitz et al., 2021) seek for a milder form, assuming the epistemic uncertainty as the standard deviation of ensemble $Q$ values. We identify the value underestimation that particularly occurs in the latter training stages and uncover its long-neglected culprit. Our findings suggest that incorporating sufficient exploitation into current exploration-driven algorithms would be a natural solution and lead to an improved algorithm.

Experience Replay (ER) (Mnih et al., 2015) boosts exploitation in off-policy RL by enabling data reuse. Recent works in prioritized replay (Schaul et al., 2015; Liu et al., 2021; Sinha et al., 2022) propose various metrics to replay or reweight important transitions more frequently, benefiting sample efficiency. We primarily implement BAC with the vanilla ER method for simplicity, yet more advanced ER techniques could be integrated for further enhancement. Outside the online RL paradigm, imitation learning (Pomerleau, 1988; Schaal, 1996; Ross et al., 2011) and offline RL algorithms (Fujimoto et al., 2019; Kumar et al., 2019; 2020; Kostrikov et al., 2021; Zhan et al., 2022) are known for their effective exploitation of provided datasets. Although the prospect of integrating these techniques to enhance online RL is attractive, offline learning is often considered overly-conservative and requires a reasonable-quality dataset for high performance (Li et al., 2022), leading to limited success in improving online learning (Niu et al., 2022). In standard online RL, we only have access to a dynamic and imperfect replay buffer, rather than a well-behaved dataset. As a result, recent efforts are mainly under a two-stage paradigm, integrating these techniques as policy pre-training for subsequent online training, such as initializing the policy with behavior cloning (Hester et al., 2018; Shah & Kumar, 2021; Wang et al., 2022a; Baker et al., 2022) or performing offline RL followed by online fine-tuning (Nair et al., 2020; Lee et al., 2022; Hansen-Estruch et al., 2023). By contrast, our work suggests a new paradigm that incorporates exploitation ingredients from offline RL to enhance pure online RL, as demonstrated in our proposed framework.

## 3 PRELIMINARIES

**Markov decision process.** We denote a discounted Markov decision process (MDP) as $\mathcal{M} = (\mathcal{S}, \mathcal{A}, P, r, \gamma)$, where $\mathcal{S}$ denotes the state space, $\mathcal{A}$ the action space, $r : \mathcal{S} \times \mathcal{A} \in [-R_{max}, R_{max}]$ the reward function, and $\gamma \in (0, 1)$ the discount factor, and $P(\cdot \mid s, a)$ stands for transition dynamics.

**Policy mixture.** During policy learning, we consider the historical policies at iteration step $k$ as a historical policy sequence $\Pi^k = \{\pi_0, \pi_1, \ldots, \pi_k\}$. Given its corresponding mixture distribution $\alpha^k$, the policy mixture $\pi_{\text{mix},k} = (\Pi^k, \alpha^k)$ is obtained by first sampling $\pi_i$ from $\alpha^k$ and then following that policy over subsequent steps. The mixture policy induces a state-action visitation density according to $d^{\pi_{\text{mix},k}}(s, a) = \sum_{i=1}^{k} \alpha_i^k d^{\pi_i}(s, a)$ (Hazan et al., 2019; Zhang et al., 2021b; Wang et al., 2022b). While the $\pi_{\text{mix},k}$ may not be stationary in general, there exists a stationary policy $\mu$ such that $d^\mu = d^{\pi_{\text{mix},k}}$.

**Off-policy actor-critic RL.** Online off-policy RL methods based on approximate dynamic programming typically utilize an action-value function $Q(s, a)$. For a given policy $\pi$, the $Q$-value can be updated by repeatedly applying a Bellman evaluation operator $\mathcal{T}$ (Sutton, 1988; Watkins, 1989):

$$\mathcal{T}Q(s, a) \triangleq r(s, a) + \gamma \mathbb{E}_{s' \sim P(\cdot|s,a)} \mathbb{E}_{a' \sim \pi(\cdot|s')}[Q(s', a')] \tag{3.1}$$

Several works under the optimism-in-face-of-uncertainty (OFU) principle could be interpreted as learning $Q$-value using a modified Bellman operator (Haarnoja et al., 2018a; Han & Sung, 2021; Moskovitz et al., 2021). We conclude them as a Bellman Exploration operator $\mathcal{T}_{explore}$ that incorporates an exploration term $\omega(s', a'|\pi)$,

$$\mathcal{T}_{explore}Q(s, a) \triangleq r(s, a) + \gamma \mathbb{E}_{s' \sim P(\cdot|s,a)} \mathbb{E}_{a' \sim \pi(\cdot|s')}[Q(s', a') - \omega(s', a'|\pi)] \tag{3.2}$$

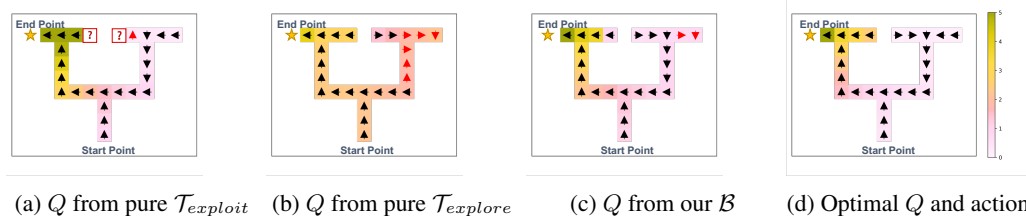

(a) $Q$ from pure $\mathcal{T}_{exploit}$    (b) $Q$ from pure $\mathcal{T}_{explore}$    (c) $Q$ from our $\mathcal{B}$    (d) Optimal $Q$ and actions

Figure 2: Comparison of different operators on a toy grid world. The agent's goal is to navigate from the bottom of the maze to the top left. The color of each square shows the learned value, red arrows reveal incorrect actions, and question marks indicate unencountered states.

## 4 EXPLOITING PAST SUCCESS FOR OFF-POLICY OPTIMIZATION

In this section, we first propose the Blended Exploration and Exploitation (BEE) operator which has good theoretical properties. Our thorough investigations highlight BEE's superior $Q$-value estimation and effectiveness in addressing the "under-exploitation" issue. Owing to its universality, we finally arrive at both model-free and model-based algorithms based on the BEE operator.

### 4.1 BLENDED EXPLOITATION AND EXPLORATION (BEE) OPERATOR

To address the under-exploitation issue, a natural idea is to extract the best-performing actions for updating the $Q$-target value. A straightforward solution might be the Bellman optimality operator, *i.e.*, $\mathcal{T}_{opt}Q(s,a) = r(s,a) + \gamma \cdot \max_{a' \in \mathcal{A}} \mathbb{E}_{s' \sim P(s'|s,a)}[Q(s',a')]$, however, it entails traversing all possible actions, being intractable in large or continuous action spaces (Kumar et al., 2019; Garg et al., 2023). In light of this, we consider the policy mixture $\mu$ induced by the replay buffer, which contains many samples and varies per policy iteration. Based on $\mu$, we introduce the Bellman Exploitation operator $\mathcal{T}_{exploit}$ to leverage the best-performing transitions from the historical policies:

$$\mathcal{T}^{\mu}_{exploit}Q(s,a) = r(s,a) + \gamma \cdot \max_{\substack{a' \in \mathcal{A} \\ \mu(a'|s') > 0}} \mathbb{E}_{s' \sim P(s'|s,a)}[Q(s',a')] \tag{4.1}$$

It yields a $Q$-value estimation that is less affected by the optimality level of the current policy. Several offline RL methods Kostrikov et al. (2021); Xu et al. (2023); Garg et al. (2023) have also focused on computing $\max Q$ constrained to the support of a pre-collected dataset for Bellman update, yet rely on a stationary behavior policy, which could be viewed as a reduced form of the $\mathcal{T}_{exploit}$ operator.

Meanwhile, to maintain the exploration optimism, we utilize the general Bellman Exploration operator in Eq.(3.2), namely,

$$\mathcal{T}^{\pi}_{explore}Q(s,a) = r(s,a) + \gamma \cdot \mathbb{E}_{s' \sim P(s'|s,a)} \mathbb{E}_{a' \sim \pi(a'|s')}[Q(s',a') - \omega(s',a'|\pi)] \tag{4.2}$$

With the Bellman Exploitation and Bellman Exploration operators, which respectively capitalize on past successes and promote the exploration of uncertain regions, we shift our focus to addressing the balance between exploitation and exploration. Here, we opt for a simple linear combination to regulate the trade-off preference, as presented below:

**Definition 4.1.** *The Blended Exploitation and Exploration (BEE) Bellman operator $\mathcal{B}$ is defined as:*

$$\mathcal{B}^{\{\mu,\pi\}}Q(s,a) = \lambda \cdot \mathcal{T}^{\mu}_{exploit}Q(s,a) + (1-\lambda) \cdot \mathcal{T}^{\pi}_{explore}Q(s,a) \tag{4.3}$$

*Here, $\mu$ is the policy mixture, $\pi$ is the current policy, and $\lambda \in (0,1)$ is a trade-off hyperparameter.*

The choice of $\lambda$ in Eq.(4.3) impacts the exploitation-exploration trade-off, as shown in Figure 2. Besides choosing a fixed number, $\lambda$ can also be autonomously and adaptively tuned with multiple methods as detailed in Appendix B.3.3. The single-operator design incurs comparable computational costs to general-purpose algorithms such as SAC (Haarnoja et al., 2018a), and is relatively lightweight compared to other methods that require training a large number of $Q$-networks to tackle the exploration-exploitation dilemma (Ciosek et al., 2019; Sun et al., 2022; Chen et al., 2021).

### 4.2 DYNAMIC PROGRAMMING PROPERTIES

For a better understanding of the BEE operator, we conduct a theoretical analysis of its dynamic programming properties in the tabular MDP setting, covering policy evaluation, policy improvement, and policy iteration. All proofs are included in Appendix A.

Figure 3: $\Delta(\mu, \pi)$ across four different tasks using an SAC agent. Blue bars correspond to positive $\Delta(\mu, \pi)$, indicating the under-exploitation stage, while orange bars represent the under-exploration stage.

**Proposition 4.2** (**Policy evaluation**). *Consider an initial* $Q_0 : \mathcal{S} \times \mathcal{A} \to \mathbb{R}$ *with* $|\mathcal{A}| < \infty$, *and define* $Q_{k+1} = \mathcal{B}^{\{\mu, \pi\}} Q_k$. *Then the sequence* $\{Q_k\}$ *converges to a fixed point* $Q^{\{\mu, \pi\}}$ *as* $k \to \infty$.

**Proposition 4.3** (**Policy improvement**). *Let* $\{\mu_k, \pi_k\}$ *be the policies at iteration* $k$, *and* $\{\mu_{k+1}, \pi_{k+1}\}$ *be the updated policies, where* $\pi_{k+1}$ *is the greedy policy of the Q-value. Then for all* $(s, a) \in \mathcal{S} \times \mathcal{A}$, $|\mathcal{A}| < \infty$, *we have* $Q^{\{\mu_{k+1}, \pi_{k+1}\}}(s, a) \geq Q^{\{\mu_k, \pi_k\}}(s, a)$.

**Proposition 4.4** (**Policy iteration**). *Assume* $|\mathcal{A}| < \infty$, *by repeating iterations of the policy evaluation and policy improvement, any initial policies converge to the optimal policies* $\{\mu^*, \pi^*\}$, *s.t.* $Q^{\{\mu^*, \pi^*\}}(s_t, a_t) \geq Q^{\{\mu', \pi'\}}(s_t, a_t), \forall \mu' \in \Pi, \pi' \in \Pi, \forall(s_t, a_t) \in \mathcal{S} \times \mathcal{A}$.

With the approximate dynamic programming properties established, our BEE operator is well-defined and flexible that could be integrated into various off-policy actor-critic algorithms.

### 4.3 Superior $Q$-value estimation using BEE operator

While being intuitively reasonable, BEE's potential benefits require further verification. In the following, we show that the BEE operator would facilitate the estimation of $Q$ and thus improve sample efficiency compared to the commonly used Bellman evaluation operator.

**Investigation on the under-exploitation stage.** As we argued in the introduction, we observe the possible under-exploitation stage after the initial under-exploration stage. To quantify the existence of under-exploitation, we compute the expected difference between the maximum $Q$-value from the historical policies and the expected $Q$-value under the current policy (considering the exploration bonus), stated as $\Delta(\mu_k, \pi_k) = \mathbb{E}_s \big[ \max_{a \sim \mu_k(\cdot|s)} Q^{\mu_k}(s, a) - \mathbb{E}_{a \sim \pi_k(\cdot|s)} [Q^{\pi_k}(s, a) - \omega(s, a|\pi_k)] \big]$ with policy mixture $\mu_k$ and current policy $\pi_k$. $\Delta(\mu_k, \pi_k)$ symbolizes the discrepancy between the value of past successes and of current policy.

A positive $\Delta(\mu_k, \pi_k)$ indicates that the value of optimal target-update actions in the replay buffer exceeds that of the actions generated by the current policy, even considering the exploration bonus. This suggests that an optimal policy derived from the replay buffer would outperform the current policy, implying a potential under-exploitation of valuable historical data. In Figure 3, we illustrate $\Delta(\mu_k, \pi_k)$ of SAC over training steps. Notably, a significant proportion of $\Delta(\mu_k, \pi_k)$ is positive in the latter training stage, suggesting that the use of the common Bellman Exploration operator $\mathcal{T}_{explore}$ does suffer from the under-exploitation issue. Further investigations on its existence and underlying reasons refer to Appendix E.2 and E.3.

**BEE mitigates the under-exploitation pitfalls.** The prevalent positive $\Delta(\mu, \pi)$ exposes the limitations of the Bellman Exploration operator $\mathcal{T}_{explore}$. The BEE operator alleviates the over-reliance on the current policy and mitigates the "under-exploitation" pitfalls by allowing the value of optimal actions in the replay buffer to be fully utilized in the $Q$-value update. To be more specific, when the $\mathcal{T}_{explore}$ operator is stuck in underestimation, the BEE operator would output a higher $Q$-value, as shown by the inequality $Q^{\{\mu_k, \pi_k\}}_{\mathcal{B}}(s, a) \geq Q^{\pi_k}_{\mathcal{T}_{explore}}(s, a) + \lambda\gamma\Delta(\mu_k, \pi_k)$. This agrees with the findings in Figure 1a, the BEE operator exhibits lower underestimation bias and faster convergence of success rate, indicating its better sample efficiency. For more results refer to Appendix E.5

**BEE exhibits no extra overestimation.** While the BEE operator seeks to alleviate underestimation, it does not incite additional overestimation. This is in contrast to prior techniques that excessively increase exploration bonuses or use optimistic estimation (Brafman & Tennenholtz, 2002; Kim et al., 2019; Pathak et al., 2019), which may distort the $Q$-value estimates and potentially cause severe overestimation (Ciosek et al., 2019). The Bellman Exploitation operator, $\mathcal{T}_{exploit}$ does not introduce

---

**Algorithm 1:** Blended Exploitation and Exploration Actor-Critic (BAC)

---

**initialize:** $Q$ networks $Q_\phi$, policy $\pi_\theta$, replay buffer $\mathcal{D}$ with $M$ samples by random policy
**for** *policy training steps* $t = 1, 2, \cdots, T$ **do**
    Sample a mini-batch of $N$ transitions $(s, a, r, s')$ from $\mathcal{D}$
    Compute $\mathcal{T}_{exploit} Q_\phi$ by Eq.(4.1)               ▷ multiple design choices available
    Compute $\mathcal{T}_{explore} Q_\phi$ by Eq.(4.2)         ▷ with a chosen exploration term $\omega(\cdot | \pi_\theta)$
    Calculate the target Q value: $\mathcal{B} Q_\phi \leftarrow \lambda \mathcal{T}_{exploit} Q_\phi + (1 - \lambda) \mathcal{T}_{explore} Q_\phi$
    **for** *each environment step* **do**
        Collect $(s, a, s', r)$ with $\pi_\theta$ from real environment; add to $\mathcal{D}$
    **for** *each gradient step* **do**
        Update $Q_\phi$ by $\min_\phi (\mathcal{B} Q_\phi - Q_\phi)^2$
        Update $\pi_\theta$ by $\max_\theta Q_\phi(s, a)$

---

artificial bonus items and instead relies solely on the policy mixture induced by the replay buffer to calculate the maximum $Q$-value. Consequently, $\mathcal{T}_{exploit}$ is grounded in real experiences.

As illustrated in Figure 4, the $Q$-value function induced by the BEE operator enjoys a lower level of overestimation and underestimation. Further, as empirically shown in Figure 1a and 2, with enhanced exploitation, the BEE operator enables faster and more accurate $Q$-value learning, thereby reducing the chains of ineffective exploration on some inferior samples, and leading to improved sample efficiency. For more results refer to Appendix E.6.

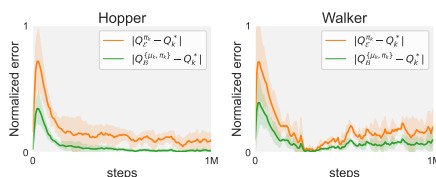

Figure 4: $Q$-value estimation error comparison. $\mathcal{T}_{explore}$ is referred to as $\mathcal{E}$ for brevity. And $Q_k^*$ is obtained practically with Monte-Carlo estimation.

## 4.4 ALGORITHMIC INSTANTIATION

We now describe two practical algorithmic instantiations based on the BEE operator $\mathcal{B}$ for both model-free and model-based RL paradigms, namely BEE Actor-Critic (BAC) and Model-Based BAC (MB-BAC), respectively. The implementation of our methods requires the specification of two main design choices: 1) a practical way to optimize the objective value on the Bellman Exploitation operator, and 2) a specific choice on the exploration term $\omega(\cdot | \pi)$ in the Bellman Exploration operator.

To effectively compute the $\max Q$-target value in Eq.(4.1) subject to the samples in the replay buffer, we utilize the in-sample learning objectives (Kostrikov et al., 2021; Garg et al., 2023; Xu et al., 2023) to learn the maximum $Q$-value over actions in the replay buffer. This treatment not only avoids the explicit computation of the policy mixture $\mu$ of replay buffer but also promotes the stability of $Q$ estimation by only extracting actions that have been previously encountered for the Bellman update.

For the exploration term $\omega(\cdot | \pi_\theta)$, numerous options have been extensively explored in prior off-policy actor-critic methods (Haarnoja et al., 2018a; Han & Sung, 2021; Eberhard et al., 2023). Here, we employ the entropy regularization term from SAC to compute $\mathcal{T}_{explore} Q_\phi(s, a)$, where actions $a'$ for target updating are extracted from $\pi_\theta$. For extensive design choices for BAC see Appendix B.3.

**Integration into Dyna-style model-based RL.** Our method could be invoked into the Dyna-style model-based RL (MBRL) framework (Sutton, 1990; 1991; Kurutach et al., 2018; Buckman et al., 2018; Luo et al., 2018). As observed in (Luo et al., 2018; Lambert et al., 2020; Ghugare et al., 2023), a better policy optimizer could potentially further enhance the algorithm's performance, this motivates us to incorporate the BEE operator in existing model-based approaches. We propose a modification to the general Dyna-style algorithm, where we replace the standard $Q$-value update rule with our BEE operator, resulting in the Model-based BAC (MB-BAC) algorithm. In contrast to previous methods that utilize SAC as policy optimization backbones (Janner et al., 2019; Lai et al., 2021; Pan et al., 2020; Ji et al., 2022), our MB-BAC algorithm treats real and model-generated data differently. It applies the $\mathcal{T}_{exploit}$ to real data $\mathcal{D}_e$, capitalizing on past successful experiences while employing the $\mathcal{T}_{explore}$ on model rollout data $\mathcal{D}_m$ to explore new possibilities. This approach enhances the effective use of valuable real data and fosters exploration in new regions of the state space. The practical

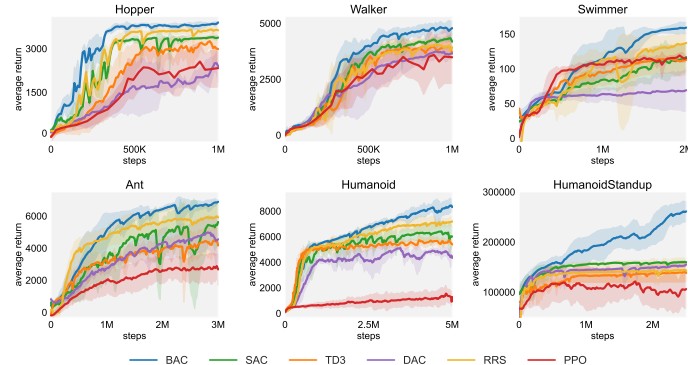

Figure 5: Training curves of BAC and five baselines on six continuous control benchmarks. Solid curves depict the mean of five trials and shaded regions correspond to the one standard deviation.

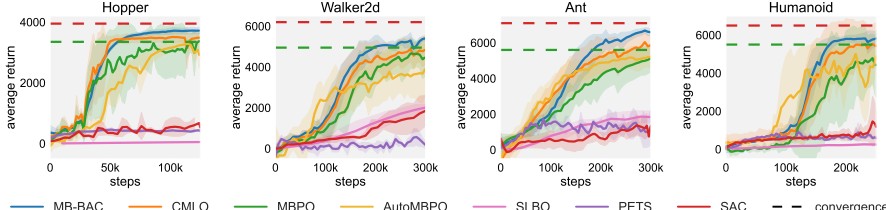

Figure 6: Training curves of MB-BAC and six baselines on four continuous control benchmarks, averaged over five trials. The dashed lines are the asymptotic performance of SAC (up to 3M) and MBPO.

implementation builds upon MBPO (Janner et al., 2019) by integrating the BAC as policy optimizer, with the pseudocode in Appendix B.2.

## 5 EXPERIMENTS

Our experimental evaluation aims to investigate the following questions: 1) How effective is the proposed BEE operator in model-based and model-free paradigms? 2) How effectively does BAC perform in failure-prone scenarios, that highlight the ability to seize serendipity and fleeting successes, particularly in various real-world tasks?

### 5.1 EVALUATION ON STANDARD CONTROL BENCHMARKS

To illustrate the effectiveness of the BEE operator across both model-based and model-free paradigms, we evaluate BAC and MB-BAC on various continuous control benchmarks. Detailed results on a total of 27 tasks from DMControl (Tunyasuvunakool et al., 2020) and Meta-World (Yu et al., 2019) benchmark suites are provided in Appendix G.

**Comparison of model-free methods.** We compare BAC to several popular model-free baselines, including: 1) SAC (Haarnoja et al., 2018a), regarded as the most popular off-policy actor-critic method; 2) TD3 (Fujimoto et al., 2018), which introduces the Double $Q$-learning trick to reduce function approximation error; 3) Diversity Actor-Critic (DAC) (Han & Sung, 2021), a variant of SAC, using a sample-aware entropy regularization instead, which is a potential choice for our $\omega(\cdot|s, a)$; 4) Random Reward Shift (RRS) (Sun et al., 2022), which learns multiple value functions (seven double-$Q$ networks) with different shifting constants for the exploration and exploitation trade-off; 5) PPO (Schulman et al., 2017), a stable on-policy algorithm that discards historical policies.

We evaluate BAC and the baselines on a set of MuJoCo (Todorov et al., 2012) continuous control tasks. BAC surpasses all baselines in terms of eventual performance, coupled with better sample efficiency, as shown in Figure 5. Notably, the HumanoidStandup task, known for its high action dimension and susceptibility to failure (Han & Sung, 2021), requires the algorithms to be able to seize and value serendipity. In this task, BAC gains a significantly better performance, with average returns up to 280,000 at 2.5M steps and 360,000 at 5M steps, which is 1.5x and 2.1x higher than the strongest baseline, respectively. This echoes the hypothesis that BAC exploits past serendipities in failure-prone environments. Trajectory visualizations in Figure 33 show that BAC agent could swiftly

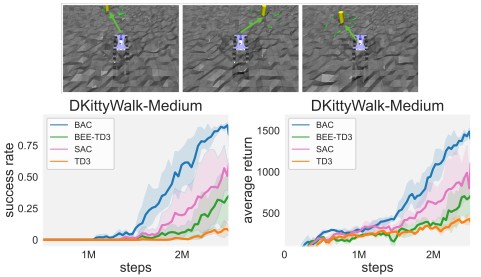

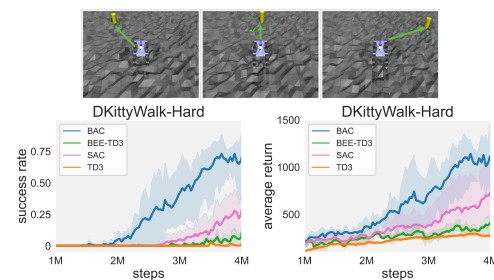

Figure 7: Success rate and average return in the DKittyWalk-Medium task.

Figure 8: Success rate and average return in the DKittyWalk-Hard task.

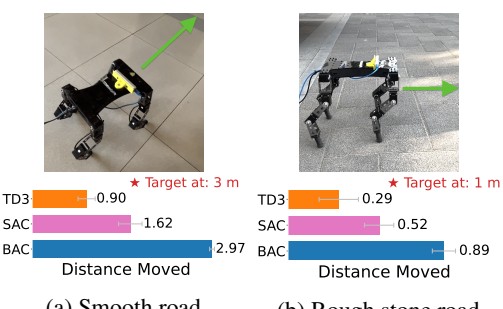

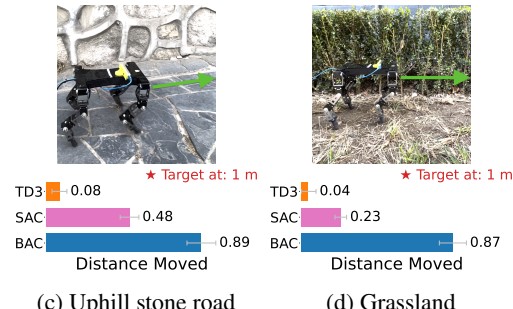

(a) Smooth road     (b) Rough stone road     (c) Uphill stone road     (d) Grassland

Figure 9: Comparisons on four challenging real-world tasks. The bar plots show how far the agent walks toward the goal for each algorithm averaged over 5 runs. For (a) and (b), we employ the policy trained in the -Medium task, and for (c) and (d) use the policy trained in the -Hard task.

reach a stable standing, while the SAC agent ends up with a wobbling kneeling posture, the DAC agent sitting on the ground, and the RRS agent rolling around.

Experiments on the more failure-prone tasks refer to Appendix F. Additionally, we integrate our BEE into the TD3 algorithm and find that the ad-hoc BEE-TD3 also outperforms the original TD3 method in 15 DMControl tasks, refer to Appendix G.1.

**Comparison of model-based methods.** We evaluate the performance of MB-BAC, which integrates the BEE operator into the MBPO algorithm, against several model-based and model-free baselines. Among the Dyna-style counterparts, MBPO (Janner et al., 2019), CMLO (Ji et al., 2022), and AutoMBPO (Lai et al., 2021) use SAC as the policy optimizer, while SLBO (Luo et al., 2018) employs TRPO (Schulman et al., 2015). PETS (Chua et al., 2018) is a planning-based method that utilizes CEM (Botev et al., 2013) as the planner. Figure 6 showcases that MB-BAC learns faster than other modern model-based RL methods and yields promising asymptotic performance compared with model-free counterparts. Moreover, the result highlights the universality of the BEE operator.

## 5.2 EVALUATION IN REAL-WORLD QUADRUPED ROBOTS WALKING TASK

We evaluate BAC on a real quadruped robot D'Kitty (Ahn et al., 2020). We follow the sim2real paradigm as in previous legged locomotion works (Agarwal et al., 2023; Ahn et al., 2020; Hwangbo et al., 2019; Tan et al., 2018) where the agent is trained in simulated environments with randomized terrains and then deployed in the real world without further training. The task is challenging, as agents are prone to falling due to fluctuating terrain. As for real-world scenarios, the D'Kitty robot is required to traverse various complex terrains, contending with unpredictable environmental factors.

Firstly, we construct two simulation task variants, DKittyWalk-Medium and DKittyWalk-Hard. The -Medium variant features a random height region of 0.07m, while the -Hard variant has a height of 0.09m, which is 1.4 times and 1.8 times higher than the base task DKittyWalkRandomDynamics, respectively. Given D'Kitty's leg length of around 0.3m when standing, navigating uneven terrain with height variations of over 0.2x to 0.3x the leg length poses a significant challenge, as a deviation of 0.02m would lead to a considerable shift in the center of gravity. Figure 7 and 8 demonstrate that BAC outperforms other algorithms in both tasks with clearer advantages. BAC achieves a success rate surpassing SAC by approximately 50%. The ad-hoc BEE-TD3 also outperforms the TD3.

More crucially, BAC achieves superior performance when deployed in the real world across various terrains, as shown in Figure 9. The policy learned in the -Medium variant is deployed on two terrains — smooth road and rough stone road, with target points positioned at distances of 3m and 1m, respectively. For more challenging terrains — uphill stone roads and grasslands, we employ the policy trained in the -Hard variant, with a target point located 1m ahead. Specifically, the BAC algorithm outperformed the TD3 and SAC agents in achieving stable movement across a variety of terrains and displaying natural gaits. In contrast, the TD3 agent prefers lower postures, such as knee walking, which makes it prone to falling on uneven terrain, while the SAC agent suffers from more oscillatory gait patterns, as shown in the supplementary videos. The empirical results also shed light on the necessity of algorithmic improvement for real-world robotics in addition to building better environments and designing informative rewards.

## 5.3 Ablation Study

**Ability to seize serendipity.** To better understand how well the BEE operator captures past well-performing actions, we conduct experiments on the DKittyWalk-Medium task. We initialize SAC and BAC with the identical $Q$ network, random policy, and replay buffer. Next, we collected 15 trajectories (2400 transitions in total) using an expert policy whose success rate is 100% and adding them to the replay buffer. Keeping all components and parameters the same as in the main experiment, we train BAC and SAC on the blended buffer harboring several successful actions. Figure 10 suggests that BAC recovers success faster than SAC, indicating its supposed ability to seize serendipity.

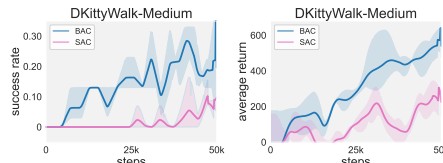

Figure 10: Comparison of the ability to seize serendipity in the DKittyWalk-Medium task. *Left*: success rate; *Right*: average return.

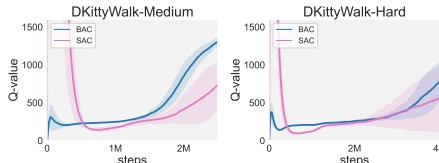

Figure 11: $Q$-value learning stability comparison. The experiments are run over 5 seeds.

**More stable $Q$-value in practice.** In failure-prone scenarios, policy performance typically experiences severe oscillation across iterations due to easily encountered failure samples from the current policy in $Q$-value update if using the Bellman evaluation operator. The $Q$-value learned by the BEE operator is less affected by the optimality level of the current policy, thus it might be expected of having better learning stability. The smaller error bar across 5 runs in Figure 11 supports it.

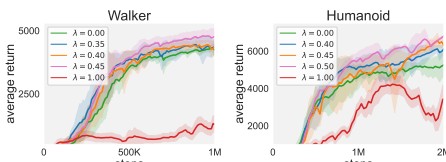

Figure 12: Parameter study on $\lambda$. The experiments are run over 4 random seeds.

**Hyperparameters.** Setting an appropriate weighted coefficient $\lambda$, BAC could balance the exploitation and exploration well. We may note that the algorithm is reduced to the online version of IQL (Kostrikov et al., 2021) for an extreme value $\lambda = 0$. According to Figure 12, and the detailed settings and hyperparameter studies in Appendix B.4, we find that a moderate choice of $\lambda$ around 0.5 is sufficient to achieve the desired performance across all 35 locomotion and manipulation tasks we have benchmarked. This underscores that BAC does not need heavy tuning for strong performance.

## 6 Conclusion

In this paper, we investigate the overlooked issue of value underestimation in off-policy actor-critic methods, which stems from "under-exploitation" in the latter training steps that hinder sample efficiency. These observations motivate us to propose the Blended Exploitation and Exploration (BEE) operator, which leverages the value of past successes to enhance $Q$-value estimation and policy learning. The proposed algorithms BAC and MB-BAC outperform both model-based and model-free methods across various continuous control tasks. Remarkably, without further training, BAC shines in real-robot tasks, emphasizing the need for improved general-purpose algorithms in real-world robotics. Finally, our work sheds light on future work on fully fusing exploitation and exploration techniques, *e.g.*, incorporating up-to-date design choices for computing $\max Q$ or exploration term, in building strong RL methods.

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

# Contents of Appendices

## A OMITTED PROOFS

**Proposition A.1** (**Policy evaluation**). *Consider an initial $Q_0 : \mathcal{S} \times \mathcal{A} \to \mathbb{R}$ with $|\mathcal{A}| < \infty$, and define $Q_{k+1} = \mathcal{B}^{\{\mu,\pi\}}Q_k$. Then the sequence $\{Q_k\}$ converges to a fixed point $Q^{\{\mu,\pi\}}$ as $k \to \infty$.*

*Proof.* First, let us show that the BEE operator $\mathcal{B}$ is a $\gamma$-contraction operator in the $\mathcal{L}_\infty$ norm.

Let $Q_1$ and $Q_2$ be two arbitrary $Q$ functions, for the Bellman Exploitation operator $\mathcal{T}_{exploit}$, since target-update actions $a'$ are extracted from $\mu$, we have that,

$$
\begin{aligned}
\|\mathcal{T}^\mu_{exploit}Q_1 - \mathcal{T}^\mu_{exploit}Q_2\|_\infty &= \max_{s,a} |(r(s,a) + \gamma\mathbb{E}_{s'} \max_{a'\sim\mu}[Q_1(s',a')]) - (r(s,a) + \gamma\mathbb{E}_{s'} \max_{a'\sim\mu}[Q_2(s',a')])| \\
&= \gamma \max_{s,a} |\mathbb{E}_{s'}[\max_{a'\sim\mu} Q_1(s',a') - \max_{a'\sim\mu} Q_2(s',a')]| \\
&\le \gamma \max_{s,a} \mathbb{E}_{s'} |\max_{a'\sim\mu} Q_1(s',a') - \max_{a'\sim\mu} Q_2(s',a')| \\
&\le \gamma \max_{s,a} \|Q_1 - Q_2\|_\infty = \gamma\|Q_1 - Q_2\|_\infty
\end{aligned}
$$

Also, for the Bellman Exploration Operator $\mathcal{T}_{explore}$, as $a' \sim \pi$, we have,

$$
\begin{aligned}
\|\mathcal{T}^\pi_{explore}Q_1 - \mathcal{T}^\pi_{explore}Q_2\|_\infty &= \max_{s,a} |\gamma\mathbb{E}_{s'}[\mathbb{E}_{a'\sim\pi}Q_1(s',a') - \gamma\mathbb{E}_{a'\sim\pi}Q_2(s',a')]| \\
&\le \gamma \max_{s,a} \mathbb{E}_{s'} |\mathbb{E}_{a'\sim\pi}Q_1(s',a') - \mathbb{E}_{a'\sim\pi}Q_2(s',a')| \\
&\le \gamma \max_{s,a} \mathbb{E}_{s'}\mathbb{E}_{a'\sim\pi} |Q_1(s',a') - Q_2(s',a')| \\
&\le \gamma \max_{s,a} \|Q_1 - Q_2\|_\infty = \gamma\|Q_1 - Q_2\|_\infty
\end{aligned}
$$

Combining the results together, we have that the BEE operator satisfies $\gamma$-contraction property:

$$
\begin{aligned}
\|\mathcal{B}^{\{\mu,\pi\}}Q_1 - \mathcal{B}^{\{\mu,\pi\}}Q_2\|_\infty &= \|\lambda(\mathcal{T}^\mu_{exploit}Q_1 - \mathcal{T}^\mu_{exploit}Q_2) + (1-\lambda)(\mathcal{T}^\pi_{explore}Q_1 - \mathcal{T}^\pi_{explore}Q_2)\|_\infty \\
&\le \lambda\|\mathcal{T}^\mu_{exploit}Q_1 - \mathcal{T}^\mu_{exploit}Q_2\|_\infty + (1-\lambda)\|\mathcal{T}^\pi_{explore}Q_1 - \mathcal{T}^\pi_{explore}Q_2\|_\infty \\
&\le \lambda\gamma\|Q_1 - Q_2\|_\infty + (1-\lambda)\gamma\|Q_1 - Q_2\|_\infty = \gamma\|Q_1 - Q_2\|_\infty
\end{aligned}
$$

we conclude that the BEE operator is a $\gamma$-contraction, which naturally leads to the conclusion that any initial Q function will converge to a unique fixed point by repeatedly applying $\mathcal{B}^{\{\mu,\pi\}}$.

$\square$

**Proposition A.2** (**Policy improvement**). *Let $\{\mu_k, \pi_k\}$ be the policies at iteration $k$, and $\{\mu_{k+1}, \pi_{k+1}\}$ be the updated policies, where $\pi_{k+1}$ is the greedy policy of the Q-value. Then for all $(s,a) \in \mathcal{S} \times \mathcal{A}$, $|\mathcal{A}| < \infty$, we have $Q^{\{\mu_{k+1},\pi_{k+1}\}}(s,a) \ge Q^{\{\mu_k,\pi_k\}}(s,a)$.*

*Proof.* At iteration $k$, $\mu_k$ denotes the policy mixture and $\pi_k$ the current policy, and the corresponding value function is $Q^{\{\mu,\pi\}}$. We firstly update the policies from $\{\mu_k, \pi_k\}$ to $\{\mu_k, \pi_{k+1}\}$, where $\pi_{k+1}$ is the greedy policy w.r.t $J_{\pi_k,\mu_k}(\mu_k, \pi)$, *i.e.*, $\pi_{k+1} = \arg\max_\pi \mathbb{E}_{a\sim\pi}[Q^{\{\mu_k,\pi_k\}}(s,a) - \omega(s,a|\pi)]$.

We commence with the proof that $Q^{\{\mu_k,\pi_{k+1}\}}(s,a) \ge Q^{\{\mu_k,\pi_k\}}(s,a)$ for all $(s,a) \in \mathcal{S} \times \mathcal{A}$. Since $\pi_{k+1} = \arg\max_\pi J_{\pi_k,\mu_k}(\mu_k, \pi)$, we have that $J_{\pi_k,\mu_k}(\mu_k, \pi_{k+1}) \ge J_{\pi_k,\mu_k}(\mu_k, \pi_k)$. Expressing $J_{\pi_k,\mu_k}(\mu_k, \pi_{k+1})$ and $J_{\pi_k,\mu_k}(\mu_k, \pi_k)$ by their definition, we have $\mathbb{E}_{a\sim\pi_{k+1}}[Q^{\{\mu_k,\pi_k\}}(s,a) - \omega(s,a|\pi_{k+1})] \ge \mathbb{E}_{a\sim\pi_k}[Q^{\{\mu_k,\pi_k\}}(s,a) - \omega(s,a|\pi_k)]$.

In a similar way to the proof of the soft policy improvement (Haarnoja et al., 2018a), we come to the following inequality:

$$
\begin{aligned}
Q^{\{\mu_k,\pi_k\}}(s_t,a_t) =& r(s_t,a_t) + \gamma\mathbb{E}_{s_{t+1}}\big\{\lambda\cdot\max_{\tilde{a}_{t+1}\sim\mu_k}Q^{\{\mu_k,\pi_k\}}(s_{t+1},\tilde{a}_{t+1})\\
&+ (1-\lambda)\cdot\mathbb{E}_{a_{t+1}\sim\pi_k}[Q^{\mu_k,\pi_k}(s_{t+1},a_{t+1})-\omega(s_{t+1},a_{t+1}|\pi_k)]\big\}\\
\leq& r(s_t,a_t) + \gamma\mathbb{E}_{s_{t+1}}\big\{\lambda\cdot\max_{\tilde{a}_{t+1}\sim\mu_k}Q^{\{\mu_k,\pi_k\}}(s_{t+1},\tilde{a}_{t+1})+\\
&(1-\lambda)\cdot\mathbb{E}_{a_{t+1}\sim\pi_{k+1}}[Q^{\{\mu_k,\pi_k\}}(s_{t+1},a_{t+1})-\omega(s_{t+1},a_{t+1}|\pi_{k+1})]\big\}\\
&\vdots\\
\leq& Q^{\{\mu_k,\pi_{k+1}\}}(s_t,a_t)
\end{aligned}
$$

Here, the inequality is obtained by repeatedly expanding $Q^{\{\mu_k,\pi_k\}}$ on the RHS through $Q^{\{\mu_k,\pi_k\}}(s,a) = r(s,a)+\gamma\mathbb{E}_{s'}\big\{\lambda\cdot\max_{\tilde{a}'\sim\mu_k}Q^{\{\mu_k,\pi_k\}}(s',\tilde{a}')+(1-\lambda)\cdot\mathbb{E}_{a'\sim\pi_k}[Q^{\{\mu_k,\pi_k\}}(s',a')-\omega(s',a'|\pi_k)]\big\}$ and applying the inequality $\mathbb{E}_{a\sim\pi_{k+1}}[Q^{\{\mu_k,\pi_k\}}(s,a)-\omega(s,a|\pi_{k+1})] \geq \mathbb{E}_{a\sim\pi_k}[Q^{\{\mu_k,\pi_k\}}(s,a)-\omega(s,a|\pi_k)]$. Finally, we arrive at convergence to $Q^{\{\mu_k,\pi_{k+1}\}}(s_t,a_t)$.

Then, we expand the historical policy sequence $\Pi_k = \{\pi_0,\pi_1,\cdots,\pi_{k-1}\}$ by adding the policy $\pi_k$, and obtain $\Pi_{k+1} = \{\pi_0,\pi_1,\cdots,\pi_k\}$. Next, we consider to prove $Q^{\{\mu_{k+1},\pi_{k+1}\}}(s,a) \geq Q^{\{\mu_k,\pi_{k+1}\}}(s,a),\forall(s,a) \in \mathcal{S}\times\mathcal{A}$. Recall that $\mu_{k+1}$ is the stationary policy mixture of $\Pi_{k+1}$, if the state-action visitation density $d^{\pi_i}(s,a) > 0, i = 0,\ldots k$, then the corresponding mixture distribution $d^\mu(s,a) > 0$, hence the support region of $\mu_k$ is a subset of the support region of $\mu_{k+1}$, i.e., $\mathrm{supp}(\mu_k) \in \mathrm{supp}(\mu_{k+1})$. Since $\max_{a\sim\mu_i}Q(s,a) = \max_{a\in\mathrm{supp}(\mu_i)}Q(s,a)$, then for any $Q : \mathcal{S}\times\mathcal{A}\to\mathbb{R}$, the following inequality can be established:

$$
\max_{a\sim\mu_{k+1}}Q(s,a) \geq \max_{a\sim\mu_k}Q(s,a),\forall s\in\mathcal{S}
$$

Hence, we expand the $Q^{\{\pi,\mu\}}$ and utilize the above inequality repeatedly, then we obtain

$$
\begin{aligned}
Q^{\{\mu_k,\pi_{k+1}\}}(s,a) =& r(s,a) + \lambda\gamma\cdot\mathbb{E}_{s'}[\max_{a'\sim\mu_k(\cdot|s')}Q^{\{\mu_k,\pi_{k+1}\}}(s',a')]\\
&+ (1-\lambda)\gamma\mathbb{E}_{s'}\mathbb{E}_{a'\sim\pi_{k+1}}[Q^{\{\mu_k,\pi_{k+1}\}}(s',a')]\\
\leq& r(s,a) + \lambda\gamma\cdot\mathbb{E}_{s'}[\max_{a'\sim\mu_{k+1}(\cdot|s')}Q^{\{\mu_k,\pi_{k+1}\}}(s',a')]\\
&+ (1-\lambda)\gamma\mathbb{E}_{s'}\mathbb{E}_{a'\sim\pi_{k+1}}[Q^{\{\mu_k,\pi_{k+1}\}}(s',a')]\\
&\vdots\\
\leq& Q^{\{\mu_{k+1},\pi_{k+1}\}}(s,a)
\end{aligned}
$$

With the inequalities of these two stages, the policy improvement property is satisfied, $Q^{\{\mu_{k+1},\pi_{k+1}\}}(s,a) \geq Q^{\{\mu_k,\pi_k\}}(s,a),\forall(s,a) \in \mathcal{S}\times\mathcal{A}, |\mathcal{A}| < \infty$.

$\square$

**Proposition A.3 (Policy iteration).** *Assume $|\mathcal{A}| < \infty$, by repeating iterations of the policy evaluation and policy improvement, any initial policies converge to the optimal policies $\{\mu^*,\pi^*\}$, s.t. $Q^{\{\mu^*,\pi^*\}}(s_t,a_t) \geq Q^{\{\mu',\pi'\}}(s_t,a_t),\forall\mu'\in\Pi,\pi'\in\Pi,\forall(s_t,a_t)\in\mathcal{S}\times\mathcal{A}.$*

*Proof.* Let $\Pi$ be the space of policy distributions and let $\{\mu_i,\pi_i\}$ be the policies at iteration $i$. By the policy improvement property in Proposition 4.3, the sequence $Q^{\{\mu_i,\pi_i\}}$ is monotonically increasing. Also, for any state-action pair $(s_t,a_t) \in \mathcal{S}\times\mathcal{A}$, each $Q^{\mu_i,\pi_i}$ is bounded due to the discount factor $\gamma$. Thus, the sequence of $\{\mu_i,\pi_i\}$ converges to some $\{\mu^*,\pi^*\}$ that are local optimum. We will still need to show that $\{\mu^*,\pi^*\}$ are indeed optimal, we assume finite MDP, as typically assumed for convergence proof in usual policy iteration (Sutton, 1988). At convergence, we get $J_{\mu^*,\pi^*}(\mu^*,\pi^*)[s] \geq J_{\mu^*,\pi^*}(\mu',\pi')[s],\forall\pi'\in\Pi,\mu'\in\Pi$. Using the same iterative augument as in the proof of Proposition 4.3, we get $Q^{\{\mu^*,\pi^*\}}(s,a) \geq Q^{\{\mu',\pi'\}}(s,a)$ for all $(s,a) \in \mathcal{S}\times\mathcal{A}$. Hence, $\{\mu^*,\pi^*\}$ are optimal in $\Pi$.

$\square$

# B  IMPLEMENTATION DETAILS AND EXTENSIVE DESIGN CHOICES

## B.1  PRIMARY IMPLEMENTATION DETAILS ON BAC

Instantiating BAC amounts to specifying two main components: the use of in-sample learning for calculating the Bellman Exploitation operator $\mathcal{T}_{exploit}$, and the application of entropy regularization in the Bellman Exploration operator $\mathcal{T}_{explore}$. Here we provide the details for our primary implementation. For a broader discussion of potential design choices and extensions refer to Section B.3.

**In-sample learning for $\mathcal{T}_{exploit}$.**  We leverage a simple and efficient approach for policy extraction using expectile regression (Kostrikov et al., 2021) to learn the value function, where only a hyper-parameter $\tau$ is introduced. Considering that some large $Q$-values potentially are a result of "lucky" samples, we introduce a state value function $V$ which approximates a high expectile of $Q(s, a)$ on the replay buffer $\mathcal{D}$. In this way, we can better account for the potential variance in $Q$-values, reducing overestimation error risk and ensuring that our algorithm is not relying solely on "lucky" samples.

To be specific, we initialize a state value $V$ network to capture the maximum of $Q$ value. Given the replay buffer $\mathcal{D}$, we can update the $V$ network by a high expectile $\tau$ of $Q(s, a)$,

$$V(s) \leftarrow \arg\min_{V} \mathbb{E}_{(s,a)\sim\mathcal{D}} \left[ |\tau - \mathbb{1}(Q(s, a) - V(s) < 0)|(Q(s, a) - V(s))\right]^2$$

Given $\tau > 0.5$, this asymmetric loss function would downweight the contributions of $Q(s, a)$ when $Q(s, a) < V(s)$ while giving more weights to larger values. If $\tau \to 1$, we have $V(s) \to \max_{a\sim\mu_k} Q(s, a)$. Hence, the target value of $\mathcal{T}_{exploit}$ can be calculated by

$$\mathcal{T}_{exploit}Q(s, a) = r(s, a) + \gamma\mathbb{E}_{s'\sim\mathcal{D}} \left[ V(s') \right].$$

**Entropy regularization in $\mathcal{T}_{explore}$.**  Based on the follow-up actions $a'$ derived from fresh policy $\pi_\theta$, we compute $\mathcal{T}_{explore}Q(s, a)$, employing the entropy regularization $\alpha\log\pi(a_t|s_t)$ from SAC (Haarnoja et al., 2018a) as the $\omega(\cdot|\pi)$. To ease the computational burden of learning a separate $V$-function for $\mathcal{T}_{explore}$, we opt to directly compute the expectation of the $Q$-value. Thus, the target value of the Bellman Exploration operator $\mathcal{T}_{exploit}$ can be calculated as follows:

$$\mathcal{T}_{explore}Q(s, a) = r(s, a) + \gamma\mathbb{E}_{s'\sim\mathcal{D}}\left[\mathbb{E}_{a'\sim\pi}Q(s', a') - \alpha\log\pi(a'|s')\right]$$

**Algorithm overview on BAC.**  The pseudocode of our proposed BAC is provided in Algorithm 2.

---

**Algorithm 2:** Primary Implementation of BEE Actor-Critic (BAC)

---

**initialize :** Q networks $Q_\phi$, V network $V_\psi$, policy $\pi_\theta$, replay buffer $\mathcal{D}$, Sample $n$ tuple from
        random policy and add to $\mathcal{D}$.
**repeat**
    **for** *each gradient step* **do**
        Sample a mini-batch of $N$ transitions $(s, a, r, s')$ from $\mathcal{D}$
        Update $V_\psi$ by $\min_\psi \mathbb{E}_{s,a}|\tau - \mathbb{1}\left(Q_\phi(s, a) < V_\psi(s)\right)|\left(Q_\phi(s, a) - V_\psi(s)\right)^2$
    **for** *each environment step* **do**
        Collect data with $\pi_\theta$ from real environment; add to $\mathcal{D}$
    **for** *each gradient step* **do**
        Compute $\mathcal{T}_{exploit}Q_\phi(s, a) \leftarrow r + \gamma\mathbb{E}_{s'}[V_\psi(s')]$
        Compute $\mathcal{T}_{explore}Q_\phi(s, a) \leftarrow r + \gamma\mathbb{E}_{s'}\mathbb{E}_{a'\sim\pi_\theta}[Q_\phi(s', a') - \alpha\log\pi_\theta(a'|s')]$
        Calculate the target Q value: $\mathcal{B}Q_\phi \leftarrow \lambda\mathcal{T}_{Exploit}Q_\phi + (1 - \lambda)\mathcal{T}_{explore}Q_\phi$
        Update $Q_\phi$ by $\min_\phi (\mathcal{B}Q_\phi - Q_\phi)^2$
        Update $\pi_\theta$ by $\max_\theta Q_\phi(s, a)$
**until** *the policy performs well in the environment*;

---

## B.2  PRIMARY IMPLEMENTATION DETAILS ON MB-BAC ALGORITHM

**Modeling and learning the dynamics models.**  We adopt the widely used model learning technique in our baseline methods (Janner et al., 2019; Lai et al., 2021; Ji et al., 2022). To be specific, MB-BAC

uses a bootstrap ensemble of dynamics models $\{\hat{f}_{\phi_1}, \hat{f}_{\phi_2}, \ldots, \hat{f}_{\phi_K}\}$. They are fitted on a shared replay buffer $\mathcal{D}_e$, with the data shuffled differently for each model in the ensemble. The objective is to optimize the Negative Log Likelihood (NLL),

$$\mathcal{L}^H(\phi) = \sum_t^H [\mu_\phi(s_t, a_t) - s_{t+1}]^T \Sigma_\phi^{-1}(s_t, a_t)[\mu_\phi(s_t, a_t) - s_{t+1}] + \log \det \Sigma_\phi(s_t, a_t).$$

The prediction for these ensemble models is, $\hat{s}_{t+1} = \frac{1}{K} \sum_{i=1}^K \hat{f}_{\phi_i}(s_t, a_t)$. More details on network settings are presented in Table 2.

**Policy optimization and model rollouts.** We employ BAC as the policy optimization oracle in MB-BAC. Using the truncated short model rollouts strategy (Janner et al., 2019; Lai et al., 2021; Pan et al., 2020; Ji et al., 2022), we generate model rollouts from the current fresh policy. In the policy evaluation step, we repeatedly apply the BEE operator to the $Q$-value. We compute the $V$-function for the Bellman Exploitation operator on the environment buffer $\mathcal{D}_e$, which contains real environment interactions collected by historical policies. And we compute the $\mathcal{T}_{explore}Q$ operation to the model buffer $\mathcal{D}_m$ generated by the current policy $\pi$.

**Algorithm overview on MB-BAC.** We give an overview of MB-BAC in Algorithm 3.

---

**Algorithm 3:** Primary Implementation of Model-based BAC (MB-BAC)

---

**initialize :** Q networks $Q_\phi$, V network $V_\psi$, policy $\pi_\theta$, ensemble models $\{\hat{f}_{\phi_1}, \hat{f}_{\phi_2}, \ldots, \hat{f}_{\phi_K}\}$,
        environment buffer $\mathcal{D}_e$ and model buffer $\mathcal{D}_m$
**repeat**
  **for** *each environment step* **do**
    Collect data with $\pi_\theta$ from real environment; add to $\mathcal{D}_e$    ▷ Interactions with real env
  **for** *each gradient step* **do**
    Train all models $\{\hat{f}_{\phi_1}, \hat{f}_{\phi_2}, \ldots, \hat{f}_{\phi_K}\}$ on $\mathcal{D}_e$    ▷ Model learning
  **for** *each model rollout step* **do**
    Perform $h$-step model rollouts using policy $\pi_\theta$; add to $\mathcal{D}_m$    ▷ Model rollouts

    ```
/* Policy optimization                                                    */
```
    Update $V_\psi$ by $\min_\psi \mathbb{E}_{s,a \sim \mathcal{D}_e} |\tau - \mathbb{1}\left(Q_\phi(s,a) < V_\psi(s)\right)| \left(Q_\phi(s,a) - V_\psi(s)\right)^2$
    Compute $\mathcal{T}_{exploit}Q_\phi(s,a) = r(s,a) + \gamma \mathbb{E}_{s' \sim \mathcal{D}_m}[V_\phi(s')]$.
    Compute $\mathcal{T}_{explore}Q_\phi(s,a) = r(s,a) + \gamma \mathbb{E}_{s' \sim \mathcal{D}_m} \mathbb{E}_{a' \sim \pi}[Q(s',a') - \alpha \log \pi_\theta(a'|s')]$.
    Calculate the target Q value: $\mathcal{B}Q_\phi \leftarrow \lambda \mathcal{T}_{exploit}Q_\phi + (1-\lambda)\mathcal{T}_{explore}Q_\phi$

  **for** *each gradient step* **do**
    Update $Q_\phi$ by $\min_\phi \left(\mathcal{B}Q_\phi - Q_\phi\right)^2$
    Update $\pi_\theta$ by $\max_\theta Q_\phi(s,a)$    ▷ Policy optimization
**until** *the policy performs well in the environment*;

---

### B.3 POSSIBLE DESIGN CHOICES AND EXTENSIONS

#### B.3.1 MORE DESIGN CHOICES ON COMPUTING $\mathcal{T}_{exploit}Q$

Towards computing $\mathcal{T}_{exploit}Q$ based on the policy mixture $\mu$, a direct solution might be using an autoencoder to model $\mu$ (Fujimoto et al., 2019; Lyu et al., 2022). Unfortunately, in the online setting, learning $\mu$ would be computationally expensive as it varies dynamically with policy iterations. In our main implementation, we use the expectile regression, an in-sample approach, for the computation of $\max Q$. Beyond this, here we introduce two other in-sample techniques that can be used to calculate $\max Q$.

**Sparse Q-learning.** Sparse Q-learning (Xu et al., 2023) considers an implicit value regularization framework by imposing a general behavior regularization term. When applied Neyman $\mathcal{X}^2$-divergence as the regularization term, the state value function can be trained by

$$V(s) \leftarrow \arg\min_V \mathbb{E}_{(s,a)\sim\mathcal{D}} \left[ \mathbb{1}\left(1 + \frac{Q(s,a) - V(s)}{2\alpha} > 0\right)\left(1 + \frac{Q(s,a) - V(s)}{2\alpha}\right)^2 + \frac{V(s)}{2\alpha} \right].$$

**Exponential Q-learning.** Similar to sparse Q-learning, exponential Q-learning (Xu et al., 2023) utilizes Reverse KL divergence as the regularization term and the state value function $V(s)$ can be updated by

$$V(s) \leftarrow \arg\min_V \mathbb{E}_{(s,a)\sim\mathcal{D}} \left[ \exp\left(\frac{Q(s,a) - V(s)}{\alpha}\right) + \frac{V(s)}{\alpha} \right].$$

Based on the state value function $V(s)$ learned by sparse Q-learning or exponential Q-learning, we can compute the $\mathcal{T}_{exploit}Q$ by,

$$\mathcal{T}_{exploit}Q(s,a) = r(s,a) + \gamma\mathbb{E}_{s'\sim\mathcal{D}}\left[V(s')\right].$$

#### B.3.2 MORE DESIGN CHOICES ON EXPLORATION TERM $\omega(\cdot|\pi)$

In our primary implementation, we adopt the widely-used entropy regularization proposed in SAC (Haarnoja et al., 2018a). Various exploration terms $\omega(s,a|\pi)$, which has been extensively explored in previous off-policy actor-critic methods (Haarnoja et al., 2018a; Fujimoto et al., 2018; Han & Sung, 2021), could be adopted in our algorithm.

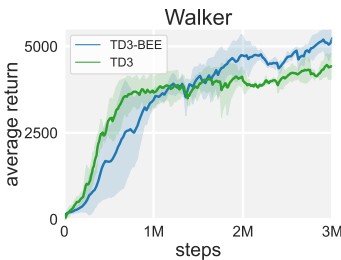

Figure 13: Ablation on the target policy smoothing regularization variant.

**Variant on target policy smoothing regularization.** Here we conduct an ablation study upon adopting the target policy smoothing regularization which introduced by TD3 (Fujimoto et al., 2018), we term the variant of our algorithm TD3-BEE. Compared to TD3, our method exhibits improvements, as demonstrated in two experiments on D'Kitty in the main paper, as well as the Walker2d-v2 experiment in Figure 13.

**Other possible extensions.** Various up-to-date advances in exploration term designs can be incorporated into our algorithm. For instance, pink noise (Eberhard et al., 2023) could be utilized to replace target policy smoothing regularization. Additionally, specially designed entropy terms, such as state or state-action occupancy entropy based on Shannon entropy (Hazan et al., 2019; Islam et al., 2019; Lee et al., 2019) or R'enyi entropy (Zhang et al., 2021a), could be considered. In certain "hard-exploration" scenarios (Aytar et al., 2018; Ecoffet et al., 2019; 2021), it may be beneficial to use specially tailored exploration terms, such as sample-aware entropy regularization (Han & Sung, 2021), particularly in sparse-reward or delayed-reward scenarios.

#### B.3.3 EXTENSIONS: AUTOMATIC ADAPTIVE $\lambda$ MECHANISMS

In our main experiments, we use a fixed $\lambda$ value for simplicity. Although the value of $\lambda$ does not fluctuate significantly in most of the environments we tested, specific scenarios may necessitate some tuning effort to find an appropriate $\lambda$.

To circumvent this $\lambda$ search, we present three possible automatic adaptation methods for $\lambda$. The first two mechanisms involve using a binary value for $\lambda$, allowing the agent to freely switch between exploration and exploitation.

- $\min(\lambda)$. The insight here is to choose the smaller of the target update values induced by the Bellman Exploration operator and Bellman Exploitation operator, which might aid in alleviating the overestimation issue and enhance learning stability. The possible drawback is that it might prefer to exclusively choose the conservative $Q$-value. We formulate this mechanism as,

$$\lambda = \mathbb{1}\left(\mathcal{T}_{exploit}Q(s,a) - \mathcal{T}_{explore}Q(s,a) \leq 0\right).$$

where $\mathbb{1}(x \leq 0)$ is an indicator function

$$\mathbb{1}(x \leq 0) = \begin{cases} 0 & x > 0, \\ 1 & x \leq 0. \end{cases}$$

- $\max(\lambda)$. This mechanism, conversely, selects the larger of the two values. This method might yield unstable results due to the influence of function approximation error. We formulate this mechanism as

$$\lambda = \mathbb{1}\left(\mathcal{T}_{exploit}Q(s,a) - \mathcal{T}_{explore}Q(s,a) \geq 0\right).$$

We also design a third mechanism for suggesting a continuous value of $\lambda$.

- $\mathrm{ada}(\lambda)$. Upon integrating new data into the replay buffer, the Bellman error variation would be small if the data is well exploited, and larger if not. Hence, when the Bellman error on the new-coming data is small, we may curtail reliance on executing $\mathcal{T}_{exploit}$ in the replay buffer and allocate more weight towards exploration. Motivated by this insight, we could adjust the value of $\lambda$ according to the Bellman error. In practice, we divide the current Bellman error $\delta_k$ by the prior Bellman error $\delta_{k-1}$ to focus more on the Bellman error caused by the introduction of new-coming data. This way, $\lambda$ can be automatically adapted during training as follows:

$$\lambda = \mathrm{clip}\left(\frac{\delta_k}{\delta_{k-1}}, 0, 1\right).$$

Here, $\mathrm{clip}(\cdot, 0, 1)$ clips the $\lambda$ by removing the value outside of the interval $[0, 1]$.

*Remark 1:* In Figure 14, we depict the learning curves of these three automatic $\lambda$ adjustment mechanisms on Walker2d and Humanoid tasks, along with the eventual performance of SAC and the primary BAC. In these two settings, the $\mathrm{ada}(\lambda)$ mechanism generally yields competitive eventual performance, while $\min(\lambda)$ and $\max(\lambda)$ are more influenced by the environment settings. For instance, in the Humanoid task, we observed that the $\min(\lambda)$ mechanism almost entirely selects 0 after 1M iterations, thus could be considered as reducing to SAC in the later stages, and its final performance matches that of SAC; however, in the Walker2d environment, $\min(\lambda)$ results in a $\lambda$ that switches more frequently.

*Remark 2:* Additionally, the third mechanism $\mathrm{ada}(\lambda)$ often yields promising results. Although it might introduce some oscillation, its advantage lies in providing guidelines for choosing $\lambda$, such as setting it to a fixed constant. As shown in Figure 15, the final fixed values of $\lambda$ chosen for these three environments fall within the range of 0.4 to 0.5.

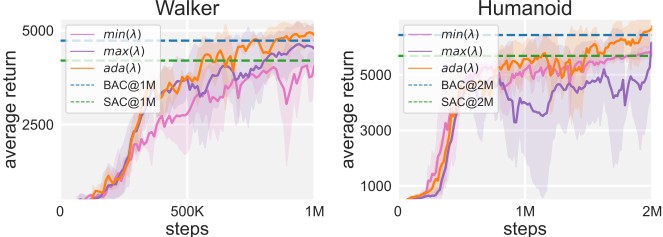
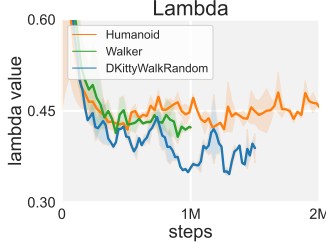

Figure 14: Learning curves with different lambda mechanisms in Walker2d and Humanoid tasks, where the dotted line indicates the eventual performance of BAC and SAC.

Figure 15: Curve of $\lambda$ with $\mathrm{ada}(\lambda)$ mechanism in different environments.

### B.4 Hyperparameter settings

**Hyperparameters for MuJoCo benchmark tasks.**    The hyperparameters used for training BAC and MB-BAC on MuJoCo benchmark tasks are outlined in Table 1 and Table 2, respectively.

In MB-BAC, we follow the hyperparameters specified in MBPO (Janner et al., 2019). The symbol "$x \rightarrow y$ over epochs $a \rightarrow b$" denotes a linear function for establishing the rollout length. That is, at epoch $t$, $f(t) = \min(\max(x + \frac{t-a}{b-a} \cdot (y - x), x), y)$. And we set $\lambda = 0.5$ and $\tau = 0.7$ for each task.

**Hyperparameters for DMControl benchmark tasks.**    We present the $\tau$ and $\lambda$ values for the 15 DMControl benchmark tasks in Table 7.

**Hyperparameters for Meta-World benchmark tasks.**    We present the $\tau$ and $\lambda$ values for the 12 Meta-World benchmark tasks in Table 8.

**Hyperparameters for other tasks.**    For other tasks, including 3 ROBEL D'DKitty tasks, 4 noisy environment tasks, and 6 sparse reward tasks, we fixed $\tau = 0.7$ and $\lambda = 0.5$ for all these tasks.

**Intuitions behind hyperparameter settings**    For practical use, $\lambda = 0.5$, $\tau = 0.7$ may suffice.

- $\lambda$: We initiated from $\lambda = 0.5$ as a balanced weight for $\mathcal{T}_{exploit}$ and $\mathcal{T}_{explore}$. Figure 14 depicts that moderate values around 0.5 obtain good performance. Besides, the automatic adaptive mechanisms we provided in Appendix B.3.3 may suffice and circumvent tuning.
- $\tau$: Our choice to primarily use 0.7 comes from the IQL paper (Kostrikov et al., 2021) which uses 0.7 for MuJoCo tasks. And $\tau = 0.7$ already suffices for expected performance, thus we mostly use 0.7 in DMControl and Meta-World tasks.

**Additional hyperparamter study**    . We provide the performance comparison of a wider set of hyperparameters in Figure 16. The results reveal that utilizing a higher $\tau = 0.9$ is not problematic and, in fact, enhances performance in ReacherHard in comparison to $\tau = 0.7$. Each BAC instance with varied hyperparameters surpasses the SAC in ReacherHard.

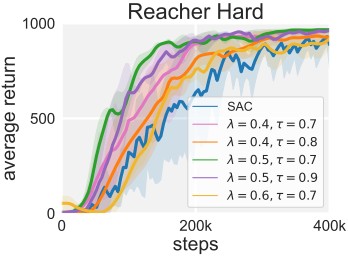

*Remark 3:* A high (e.g., 0.9) $\tau$ may not be problematic in the online setting. This differs from offline RL. In the offline setting, the peak distributions will occur in various quantiles for different datasets, thus an unsuitable $\tau$ may cause erroneous estimation. However, in online settings, ongoing interactions could enrich peak data. As policy improves, the replay buffer accumulates high-value data, thus reducing sensitivity to $\tau$.

Figure 16: Learnings curves of BAC with a wider set of hyperparameters on ReacherHard.

**Additional results with unified hyperparameter setting.**    The parameter set $\lambda = 0.5$, $\tau = 0.7$ is sufficient for practical use. Previously, in 41 of 50 experiments spanning diverse tasks and benchmarks (MuJoCo, DMControl, MetaWorld, Robel, Panda-gym), this hyperparameter set consistently achieved SOTA performance as reported in our paper. The remaining 9 tasks are only slightly tuned, as we find that BAC can achieve even better performance. We demonstrate the results of BAC with $\lambda = 0.5$, $\tau = 0.7$ on the left 9 tasks in Figure 17 and 18. In a nutshell, using a unified setting of hyperparameters is sufficient for strong performance.

### B.5 Computing infrastructure and computational time

Table 3 presents the computing infrastructure used for training our algorithm BAC and MB-BAC on benchmark tasks, along with the corresponding computational time.

Compared to SAC, the total training time of BAC only increased by around 8% for Humanoid (1.19 H for 5M steps). Thus. we believe the additional costs are acceptable. Further, for practical use, BAC requires fewer interactions for similar performance, which may lower the needed computation time.

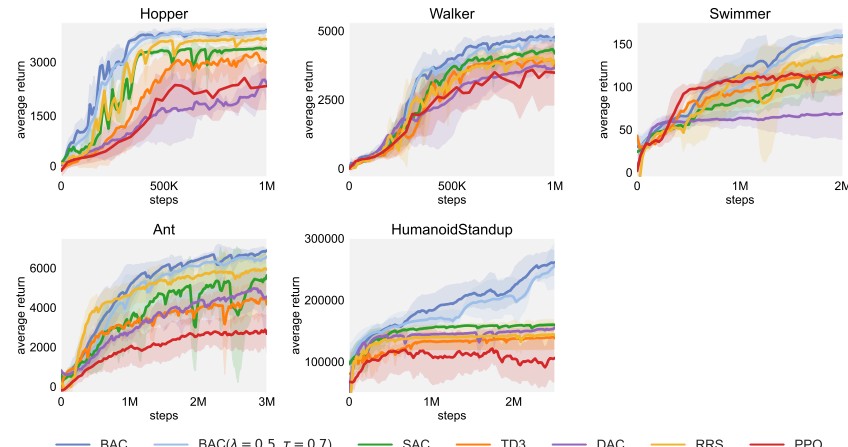

Figure 17: Training curves of BAC with (with $\lambda = 0.5, \tau = 0.7$ and the hyperparameters in the paper) on MuJoCo tasks. Solid curves depict the mean of five trials and shaded regions correspond to the one standard deviation.

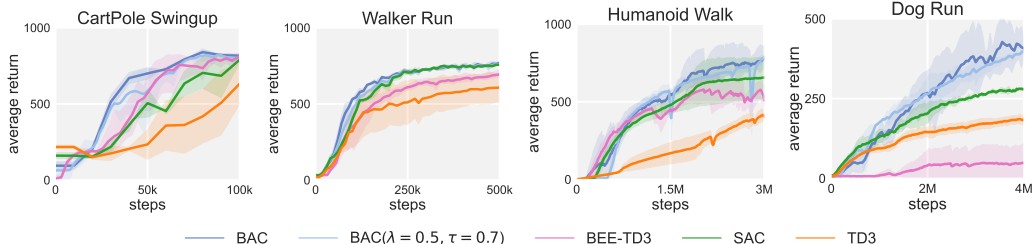

Figure 18: Training curves of BAC with (with $\lambda = 0.5, \tau = 0.7$ and the hyperparameters in the paper) on DMControl tasks, averaged over five trials.

Table 1: Hyperparameter settings for BAC in MuJoCo benchmark tasks.

|  | Hopper | Walker2d | Swimmer | Ant | Humanoid | HumanoidStandup |
|---|---|---|---|---|---|---|
| $Q$-value network | \multicolumn{6}{c}{MLP with hidden size 512} ||||||
| $V$-value network | \multicolumn{6}{c}{MLP with hidden size 512} ||||||
| policy network | \multicolumn{6}{c}{Gaussian MLP with hidden size 512} ||||||
| discounted factor | \multicolumn{6}{c}{0.99} ||||||
| soft update factor | \multicolumn{6}{c}{0.005} ||||||
| learning rate | \multicolumn{6}{c}{0.0003} ||||||
| batch size | \multicolumn{6}{c}{512} ||||||
| policy updates per step | \multicolumn{6}{c}{1} ||||||
| value updates per step | \multicolumn{6}{c}{1} ||||||
| $\lambda$ | 0.4 | 0.45 | 0.5 | 0.5 | 0.5 | 0.5 |
| $\tau$ | 0.8 | 0.8 | 0.8 | 0.8 | 0.7 | 0.8 |

Table 2: Hyperparameter settings for MB-BAC in MuJoCo benchmark tasks.

|  | Hopper | Walker2d | Ant | Humanoid |
|---|---|---|---|---|
| dynamical models network | Gaussian MLP with 4 hidden layers of size 200 | | | |
| ensemble size | 5 | | | |
| model rollouts per policy update | 400 | | | |
| rollout schedule | $1 \rightarrow 15$ over epochs $20 \rightarrow 100$ | 1 | $1 \rightarrow 25$ over epochs $20 \rightarrow 100$ | $1 \rightarrow 25$ over epochs $20 \rightarrow 300$ |
| policy network | Gaussian with hidden size 512 | | Gaussian with hidden size 1024 | |
| policy updates per step | 40 | 20 | 20 | 20 |

Table 3: Computing infrastructure and the computational time for MuJoCo benchmark tasks.

|  | Hopper | Walker | Swimmer | Ant | HumanoidStandup | Humanoid |
|---|---|---|---|---|---|---|
| CPU | AMD EPYC 7763 64-Core Processor (256 threads) | | | | | |
| GPU | NVIDIA GeForce RTX 3090 $\times$ 4 | | | | | |
| **BAC** computation time in hours | 3.21 | 3.48 | 6.56 | 13.47 | 11.65 | 16.57 |
| **MB-BAC** computation time in hours | 18.35 | 19.51 | - | 27.57 | - | 30.86 |

## C   ENVIRONMENT SETUP

We evaluate the BEE operator across 35 diverse continuous control tasks, spanning **MuJoCo, Robel, DMControl, and Meta-World**. It excels in both locomotion and manipulation tasks. Besides, we conduct experiments on 4 noisy environments and 6 sparse reward tasks to further showcase the effectiveness of BEE operator. As a versatile plugin, it seamlessly enhances performance with various policy optimization methods, shining in model-based and model-free paradigms.

We also validate BAC using a **cost-effective D'Kitty robot** to navigate various complex terrains and finally reach goal points and desired postures. The 4 real-world quadruped locomotion tasks highlight BAC's effectiveness in real-world scenarios.

★ Visualizations of these tasks are provided in Figure 19.

### C.1   ENVIRONMENT SETUP FOR EVALUATING BAC

**MuJoCo benchmark tasks.**   We benchmark BAC on six continuous control tasks in OpenAI Gym (Brockman et al., 2016) with the MuJoCo (Todorov et al., 2012) physics simulator, including Hopper, Walker2d, Swimmer, Ant, Humanoid, HumanoidStandup, using their standard versions.

**D'Kitty simulated tasks.**   ROBEL (Ahn et al., 2020) is an open-source platform of cost-effective robots designed for real-world reinforcement learning. The D'Kitty robot, with four legs, specializes in agile-legged locomotion tasks. ROBEL platform provides a set of continuous control benchmark tasks for the D'Kitty robot.

To construct more challenging locomotion tasks, we modify the base task DKittyWalkRandomDynamics by increasing terrain unevenness.

Details for the base task (DKittyWalkRandomDynamics):

- Task: D'Kitty moves from an initial position $p_{t,kitty}$ to a desired one $p_{goal}$ while maintaining balance and facing direction.
- Setting: Randomized joint parameters, damping, friction, and terrain with heights up to $0.05$m.
- Reward function: The reward function contains five parts, the upright standing reward $r_{t,upright}$, the distance penalty $d_{t,goal} = \|p_{goal} - p_{t,kitty}\|_2$, the heading alignment $h_{t,goal} = R_{y,t,kitty}(p_{goal} - p_{t,kitty})/d_{t,goal}$, a small task success bonus $r_{bonus\_small}$ and a big task success bonus $r_{bonus\_big}$. Thus, the reward function is defined as

$$r_t = r_{t,upright} - 4d_{t,goal} + 2h_{t,goal} + r_{bonus\_small} + r_{bonus\_big}.$$

- Success indicator: The success is defined by meeting distance and orientation criteria. The formulation of success indicator is:

$$\phi_{se}(\pi) = \mathbb{E}_{\tau \sim \pi} \left[ \mathbb{1}\left(d_{T,goal}^{(\tau)} < 0.5\right) * \mathbb{1}\left(u_{T,kitty}^{(\tau)} > \cos(25°)\right)\right].$$

The "Success Rate" in our experiments refers to the success percentage over 10 runs.

We conduct experiments on two more challenging variants:

- *DKittyWalk-Hard*: the randomized height field is generated with heights up to 0.07m.
- *DKittyWalk-Medium*: the randomized height field is generated with heights up to 0.09m.

**D'Kitty real-world tasks.**   Our real-world validation experiments are performed using a cost-effective D'Kitty robot. D'Kitty Ahn et al. (2020) is a twelve-DOF quadruped robot capable of agile locomotion. It consists of four identical legs mounted on a square base. Its feet are 3D-printed parts with rubber ends.

The D'Kitty robot is required to traverse various complex terrains, contending with unpredictable environmental factors, and finally reach a target point. We evaluate BAC and baseline methods on four different terrains: *smooth road* (with a target point at 3m), *rough stone road* (target point at 1m), *uphill stone road* (target point at 1m), and *grassland* (target point at 1m).

**DMControl tasks.** The DeepMind Control Suite (DMControl) (Tunyasuvunakool et al., 2020), provides a set of continuous control tasks with standardized structures and interpretable rewards. We evaluate BAC BEE-TD3, SAC, and TD3 on 15 diverse benchmark tasks from DMControl, including challenging high-dimensional tasks like Humanoid Walk, Humanoid Run, DogWalk, and DogRun.

**Meta-World tasks.** Meta-World (Yu et al., 2019) provides a suite of simulated manipulation tasks with everyday objects, all of which are contained in a shared, tabletop environment with a simulated Sawyer arm. We evaluate BAC in 12 individual Meta-World tasks. Note that we conduct experiments on the goal-conditioned versions of the tasks from Meta-World-v2, which are considered harder than the single-goal variant often used in other works.

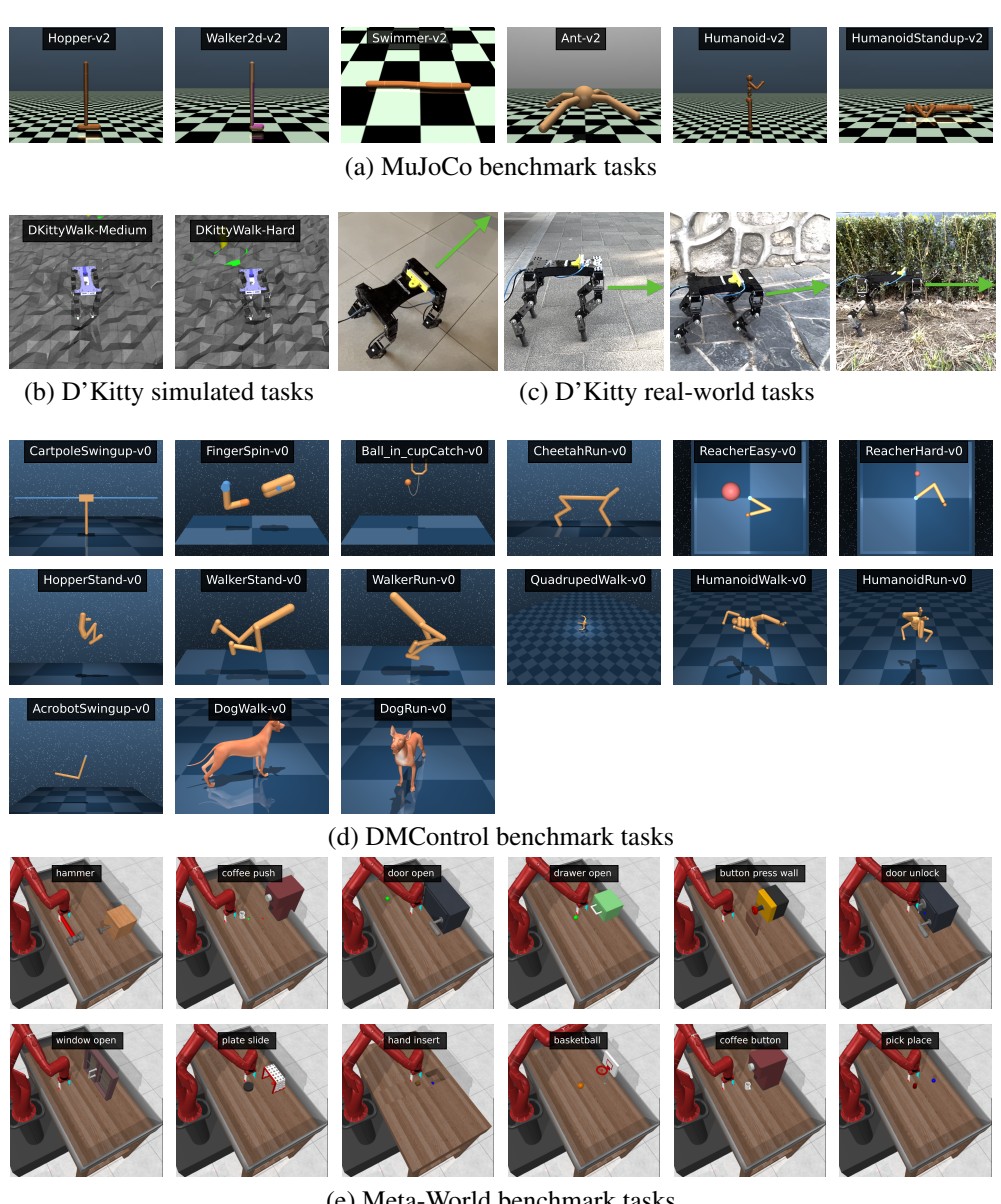

(a) MuJoCo benchmark tasks

(b) D'Kitty simulated tasks    (c) D'Kitty real-world tasks

(d) DMControl benchmark tasks

(e) Meta-World benchmark tasks

Figure 19: Visualization of **35 simulated tasks** and **4 real-world robot tasks**.

Table 4: Overview on environment settings for MB-BAC and model-based baselines. Here, $\theta_t$ denotes the joint angle and $z_t$ denotes the height at time $t$.

| | State Space Dimension | Action Space Dimension | Horizon | Terminal Function |
|---|---|---|---|---|
| Hopper-v2 | 11 | 3 | 1000 | $z_t \leq 0.7$ or $\theta_t \geq 0.2$ |
| Walker2d-v2 | 17 | 6 | 1000 | $z_t \geq 2.0$ or $z_t \leq 0.8$ or $\theta_t \leq -1.0$ or $\theta_t \geq 1.0$ |
| Ant-v2 | 27 | 8 | 1000 | $z_t < 0.2$ or $z_t > 1.0$ |
| Humanoid-v2 | 45 | 17 | 1000 | $z_t < 1.0$ or $z_t > 2.0$ |

## C.2 ENVIRONMENT SETUP FOR EVALUATING MB-BAC

We evaluate MB-BAC and its counterparts on four continuous control tasks in MuJoCo (Todorov et al., 2012). To ensure a fair comparison, we follow the same settings as our model-based baselines (MBPO (Janner et al., 2019), AutoMBPO (Lai et al., 2021), CMLO (Ji et al., 2022)), in which observations are truncated. The details of the experimental environments are provided in Table 4.

## D  BASELINES IMPLEMENTATION

**Model-free RL algorithms.** We compare with five popular model-free baselines, Soft Actor-Critic (SAC) (Haarnoja et al., 2018a), Diversity Actor-Critic (DAC) (Han & Sung, 2021), Random Reward Shift (RRS) (Sun et al., 2022), Twin Delayed DDPG (TD3) (Fujimoto et al., 2018), Proximal Policy Optimization (PPO) (Schulman et al., 2017). For RRS, we use the RRS-7 0.5 version as it provides better performance across diverse environments compared to other alternatives (RRS-3 0.5, RRS-3 1.0, RRS-7 1.0). For MuJoCo tasks, the hyperparameters of DAC, RRS, TD3, and PPO are kept the same as the authors' implementations. We list the hyperparameters of TD3 in Table 6. Note that we mostly follow the implementation of the original paper but improve upon certain hyperparameter choices for DMControl and Meta-World tasks.

The implementation of SAC is based on the open-source repo (pranz24 (2018), MIT License). And we use automating entropy adjustment (Haarnoja et al., 2018b) for automatic $\alpha$ tuning. On MuJoCo benchmarks, we retain other parameters as used by the authors (Haarnoja et al., 2018a). On DM-Control benchmarks, we followed the SAC hyperparameters suggested by TD-MPC paper (Hansen et al., 2022). On Meta-World benchmarks, we followed the SAC hyperparameters suggested by Meta-World paper (Yu et al., 2019). We list the hyperparameters of SAC in Table 5.

**Model-based RL algorithms.** As for model-based methods, we compare with four state-of-the-art model-based algorithms, MBPO (Janner et al., 2019), SLBO (Luo et al., 2018), CMLO (Ji et al., 2022), AutoMBPO (Lai et al., 2021). The implementation of SLBO is taken from an open-source MBRL benchmark (Wang et al., 2019), while MBPO is implemented based on the `MBRL-LIB` toolbox (Pineda et al., 2021). To facilitate a fair comparison, MB-BAC and MBPO are run with identical network architectures and training configurations as specified by `MBRL-LIB`.

Table 5: Hyperparameter settings for SAC in MuJoCo and DMControl, Meta-World benchmark tasks.

| | MuJoCo | DMControl | Meta-World |
|---|---|---|---|
| optimizer for $Q$ | | Adam($\beta_1$=0.9, $\beta_2$=0.999) | |
| optimizer for $\alpha$ | | Adam($\beta_1$=0.5, $\beta_2$=0.999) | |
| learning rate | $3 \times 10^{-4}$ | $1 \times 10^{-4}$ (otherwise) $3 \times 10^{-4}$ (Dog) | $3 \times 10^{-4}$ |
| discount ($\gamma$) | | 0.99 | |
| number of hidden units per layer | 256 | 1024 | 256 |
| number of samples per minibatch | 256 | 512 (otherwise) 2048 (Dog) | 500 |
| target smoothing coefficient ($\tau$) | | 0.005 | |
| target update interval | 1 | 2 | 1 |
| gradient steps | | 1 | |

Table 6: Hyperparameter settings for TD3 in MuJoCo and DMControl, Meta-World benchmark tasks.

| | MuJoCo | DMControl | Meta-World |
|---|---|---|---|
| optimizer for $Q$ | | Adam($\beta_1$=0.9, $\beta_2$=0.999) | |
| exploration noise | | $\mathcal{N}(0, 0.1)$ | |
| learning rate | $3 \times 10^{-4}$ | $1 \times 10^{-4}$ (otherwise) $3 \times 10^{-4}$ (Dog) | $3 \times 10^{-4}$ |
| discount ($\gamma$) | | 0.99 | |
| hidden layers | (400, 300) | (512, 512) | (512, 512) |
| number of samples per minibatch | 100 | 256 (otherwise) 512 (Dog) | 256 |
| target smoothing coefficient ($\tau$) | | 0.005 | |
| target update interval | | 1 | |

# E INVESTIGATIONS ON THE UNDERESTIMATION ISSUE IN THE UNDER-EXPLOITATION STAGE

The underestimation issue from under-exploitation matters. While prior work focus more on reducing overestimation, our work shows that mitigating underestimation itself may improve both performance and sample efficiency. Let's delve deeper.

## E.1 WHY UNDERESTIMATION AND UNDER-EXPLOITATION MATTERS?

Underestimation in the under-exploitation stage would negatively impact $Q$-value estimation. Underestimating Q-values of $(s, a)$ due to suboptimal current policy successors, ignoring high-value replay buffer successors, hampers reselection of $(s, a)$. Two issues might arise,

- **Reduce sample efficiency**: The agent would require more samples to re-encounter such $(s, a)$.

- **Hinder policy learning**: Misleading $Q$ may trap the policy in ineffective exploration. The issue is exacerbated in failure-prone scenarios where high-value tuples are serendipities and policy performance oscillates.

## E.2 CULPRITS OF THE UNDER-EXPLOITATION CIRCUMSTANCE

Many Actor-Critic algorithms commonly encounter the circumstance: the actions sampled from the current policy $\pi$ fall short of the optimal ones stored in the replay buffer $\mathcal{D}$. It underscores a prevalent challenge: the insufficient utilization of the potentially superior data collected from historical policies, that why we term it as *under-exploitation*. Several factors contribute to this circumstance:

**Exploration bias**: Exploration bias often leads to the overestimation of Q-values, promoting policy exploration of suboptimal actions.

**AC framework nature**: Consideration of the iterative update nature of the Actor-Critic (AC) framework also brings two additional dimensions into play:

- $Q$-**value estimation bias**: During the training process, either underestimation or overestimation is inevitable. In other words, the true $Q$-value of the sampled actions from the current policy might be lower than some actions in the replay buffer.

- **Suboptimal policy update**: Ideally, each new policy should be the maximizer of the current $Q$ to ensure policy improvement. However, obtaining such an optimal policy w.r.t the current $Q$ function is practically unattainable with a few policy gradient updates.

## E.3 THE EXISTENCE OF UNDER-EXPLOITATION STAGE

**Existence without exploration bias.** The existence of better actions in the replay buffer stems not solely from entropy term. It also attributes to the particulars of the optimization in AC, as obtaining an optimal policy w.r.t the current Q value is practically unattainable with a few policy gradient updates. Many off-policy AC methods, relying solely on current policy for $Q$-value updates, may face under-exploitation issue.

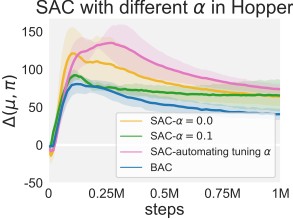
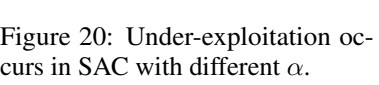
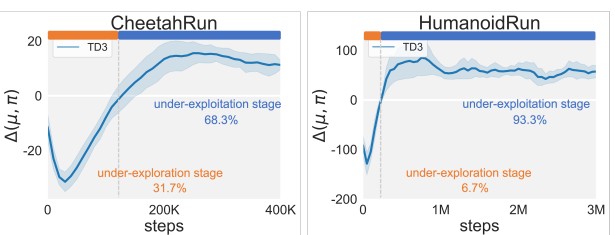

Figure 20: Under-exploitation occurs in SAC with different $\alpha$.

Figure 21: Visualization of $\Delta(\mu, \pi)$ on TD3 agent. Positive $\Delta(\mu, \pi)$ indicates the under-exploitation stage.

Figure 20 illustrates the under-exploitation occurs in SAC with varying $\alpha$. Notably, under-exploitation is observed even in SAC instances with $\alpha = 0$, indicating the presence of under-exploitation even when there is no exploration bias. BAC mitigates under-exploitation pitfalls more, even equipped with an exploration term, when compared to the SAC instance with $\alpha = 0$.

**Existence in TD3.** Under-exploitation exists in many off-policy algorithms, not limited to SAC. Figure 21 shows that TD3 also encounters under-exploitation stages during training.

**Existence in many scenarios.** Here we provide more results on the existence of under-exploitation stage, as shown in Figure 22, that in various scenarios, positive $\Delta(\mu_k, \pi_k)$ occupies a significantly larger portion than negative $\Delta(\mu_k, \pi_k)$, indicating that the common Bellman Exploration operator $\mathcal{T}_{explore}$ suffers from under-exploitation stages for a prolonged period of time.

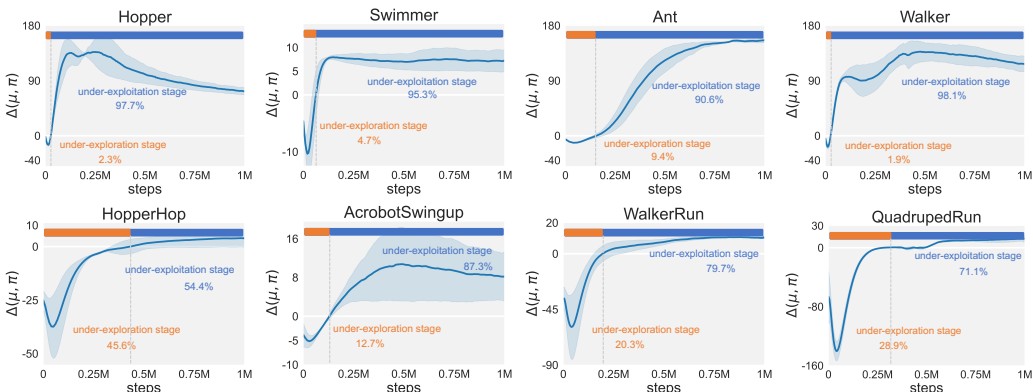

Figure 22: Visulization on under-exploitation stage on eight environments across MuJoCo and DMControl benchmark tasks based on the SAC agent.

**Positive $\Delta(\mu, \pi)$ during later training stages.** From the visualization figures above, we often observe a positive $\Delta(\mu, \pi)$ during later training stages, indicating that the initial under-exploration stage is often followed by a subsequent under-exploitation stage. To give more insights,

- In the early training stages, the policy $\pi$ performs poorly and possibly more randomly, resulting in 1) low-reward samples in the replay buffer with corresponding low $Q$ values; 2) the exploration bonus improves the expected $Q$-value of the current policy.

- As training progresses, and the agent begins to solve the task, better actions than those generated by $\pi$ may appear in the replay buffer. It is partially attributed to the iterative update nature of the Actor-Critic (AC) framework as discussed in Appendix E.2, which may lead to the existence of inferior actions after policy updates compared to the optimal ones in the replay buffer.

### E.4 UNDERESTIMATION ISSUE IN UNDER-EXPLOITATION STAGE

Previous works suggest that double-Q-trick may cause underestimation (Fujimoto et al., 2018; Moskovitz et al., 2021). We identify that the underestimation problem also occurs in many off-policy Actor-Critic (AC) algorithms independent of this trick. In this section, we investigate the causes of underestimation in the AC framework, irrespective of the double-Q trick's application. We also empirically show that various off-policy AC algorithms, with or without the double-Q trick, are prone to underestimation issues in many tasks.

**The optimization procedure of AC framework can also contribute underestimation.** Ideally, the Bellman update needs to solve $Q(s, a) \leftarrow r(s, a) + \gamma \mathbb{E}_s[\max_a Q(s, a)]$. However, as $\max_a Q(s, a)$ operations are often impractical to calculate, so in the AC framework, we typically iteratively evaluate target Q-value as $\mathbb{E}_\pi[Q(s, a)]$, while implicitly conducting the max-Q operation in a separate policy improvement step to learn policy $\pi$.

Note that the ideal $\pi = \arg\max_{a\sim\pi} Q(s,a)$ is not possible to achieve practically within only a few policy gradient updates [2]. Hence, the actual target value used in AC Bellman update $\mathbb{E}_{s,a\sim\pi}Q(s,a)$ can have a high chance to be smaller than $\mathbb{E}_s[\max_a Q(s,a)]$, causing underestimation. In other words, the non-optimal current policy in the AC framework can also contribute to underestimation.

The existence of more optimal actions in the replay buffer than generated by current policy further supports the actual gap to the optima. We identify that underestimation particularly occurs in the latter training stage, where we see a notable shortfall in the exploitation of the more optimal actions in the replay buffer. Thus, exploiting the more optimal actions in the replay buffer to bootstrap $Q$ would shorten the gap to the optima, hence mitigates underestimation.

**Underestimation issue of TD3.** AC algorithms are susceptible to the underestimation issue, as shown in Figure 1b. To further illustrate this issue, we quantify the $Q$-value estimation gap of TD3 and BEE-TD3 in the DKittyWalkRandomDynamics task. The gap in Q estimation is evaluated by comparing the TD3/BEE-TD3's $Q$-values and the Monte-Carlo $Q$ estimates using the trajectories in the replay buffer.

As shown in Figure 23, TD3 also experiences under-estimation pitfalls during the later stages of training. Notably, we observe that the BEE operator help to mitigate this underestimation issue and finally bene-fits performance.

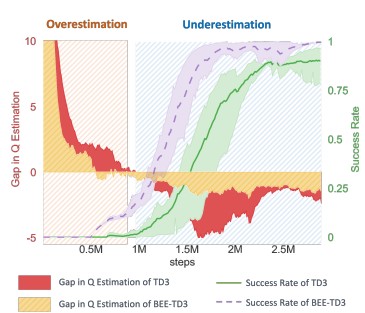

Figure 23: TD3 is also prone to underestimation pitfalls in the latter stage of training.

**Underestimation issue without doule-Q-trick.** While the double-Q-trick Fujimoto et al. (2018); Haarnoja et al. (2018a) might lower Q-value esti-mation Moskovitz et al. (2021), the underestimation issue is not solely an outcome of the double-Q-trick. To recap, the underestimation issue primarily arises from the optimization procedure of the Actor-Critic (AC) framework. Thus, this issue occurs even in the SAC and TD3 instances without the double-Q-trick, as shown in the visualization on 8 different tasks, please refer to Figure 24.

### E.5 BEE MITIGATES THE UNDER-EXPLOITATION PITFALLS

As detailed in Section 4.3, the BEE operator mitigates under-exploitation pitfalls by reducing its reliance on the current policy and fully utilizing the value of optimal actions stored in the replay buffer during the $Q$-value update.

To further illustrate this, we visualize the $\Delta(\mu, \pi)$ metric for both SAC and BAC agents in Hopper and Swimmer environments. As shown in Figure 25, BAC improves the metric $\Delta(\mu, \pi)$ more towards 0, indicating its ability to learn more accurate $Q$-values. BEE operator prevents a suboptimal current policy from diminishing the value of these actions. Thus, BAC has a higher likelihood of re-encountering these high-valued actions used for computing target $Q$-value, effectively mitigating under-exploitation pitfalls.

### E.6 BEE EXHIBITS NO EXTRA OVERESTIMATION

While the BEE operator seeks to alleviate underestimation, it does not incite additional overestimation. In Figure 4, we observe that the Q-value function induced by the BEE operator enjoys a lower level of overestimation and underestimation. Here we consider an extreme suitation, $\lambda = 1$. We plot Q-estimation-error under $\lambda = 1$ in Figure 26, and find that it does not cause overestimation.

Actually, $\mathcal{T}^{\mu}_{exploit}$, the reduced form BEE operator when $\lambda = 1$, relies on real experience and may lead to conservative estimation. To give more insights, online learning's dynamic replay memory could be treated as a static dataset at a specific time step. Then in practice, the Bellman exploitation operator $\mathcal{T}^{\mu}_{exploit}$ could be obtained by several effective techniques from offline RL. The pessimistic

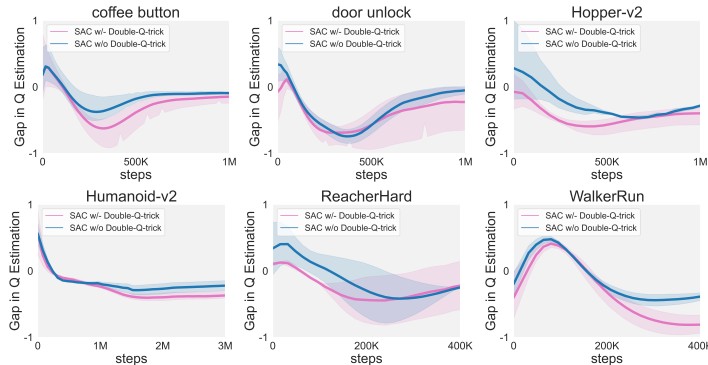

(a) SAC with or without double-Q-trick faces underestimation in the latter stage of training.

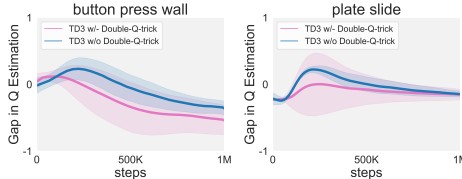

(b) TD3 with or without double-Q-trick faces underestimation in the latter stage of training.

Figure 24: Off-policy Actor-Critic algorithms with or without double-Q-trick are prone to underestimation pitfalls.

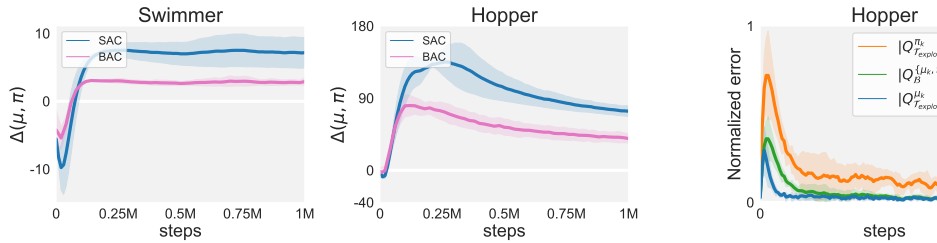

Figure 25: Visualization of $\Delta(\mu, \pi)$ with SAC and BAC agent in Hopper and Swimmer tasks. Here, BAC improves the metric more towards 0 compared to SAC.

Figure 26: $Q$-value estimation error of different operators.

treatments in offline RL penalize overestimation heavily. Thus a pure exploitation operator practically even might help to reduce overestimation.

## F EFFECTIVENESS IN FAILURE-PRONE SCENARIOS

In our main paper, we have shown the effectiveness of the BEE operator in terms of **ability to seize serendipity** and **more stable $Q$-value in practice**. Here, we investigate the superiority of the BEE operator in terms of **ability to seize serendipity** and **effectiveness in noisy environments**.

### F.1 THE ABILITY TO COUNTERACT FAILURE

The BEE operator can not only grasp success but also **counteract failure**. Here, we conduct some extreme experiments to show it. We simultaneously train SAC and BAC, and at 100k steps, both have reached a certain level of performance. This suggests that there already exists several high-value (successful) samples in the replay buffer. At this point, we abruptly apply a massive perturbation to the policy and value networks (*i.e.*, at 100k steps, we substitute the current policy with a random one and reinitialize the value networks). Keep other components the same, we continue the training. This setup is a magnification of a situation often seen in failure-prone scenarios: the agent is prone to performance drop, which consequently disrupts the $Q$ value estimate and necessitates additional sampling for recovery, thus forming a stark gap in the learning curve.

As shown in Figure 27, we can observe that the degree of performance drop in BAC after 100k steps is significantly less than that in SAC, coupled with a faster recovery speed, demonstrating its better resilience against failure. This capability possibly stems from the fact that the learned $Q$ value by the BEE operator is less influenced by the optimal level of the current policy.

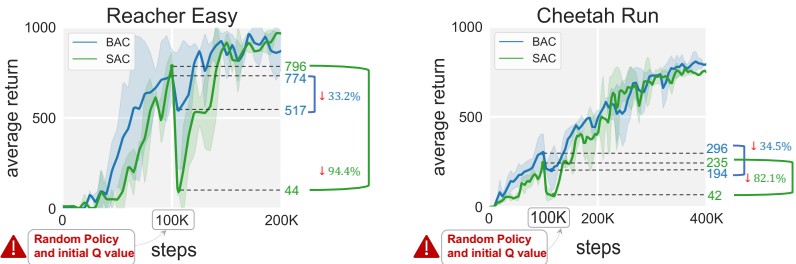

Figure 27: Comparison of the ability to counteract failure. BAC exhibits less performance drop (33.2% in ReachEasy and 34.5% in CheetahRun) and faster recovery.

### F.2 EFFECTIVENESS IN NOISY ENVIRONMENTS

We conduct experiments in noisy environments to investigate the robustness of the BEE operator. Noisy environments are created by adding Gaussian noise to the agent's action at each step. Specifically, the environment executes $a' = a + \text{WN}(\sigma^2)$ as the action, where $a$ is the agent's original action, and $\text{WN}(\sigma^2)$ is an unobserved Gaussian white noise with a standard deviation of $\sigma$.

Despite the noisy settings that can destabilize $Q$ values and impede sample efficiency, as shown in Figure 28, BAC demonstrates desirable performance, even outperforming SAC more significantly than in noise-free environments.

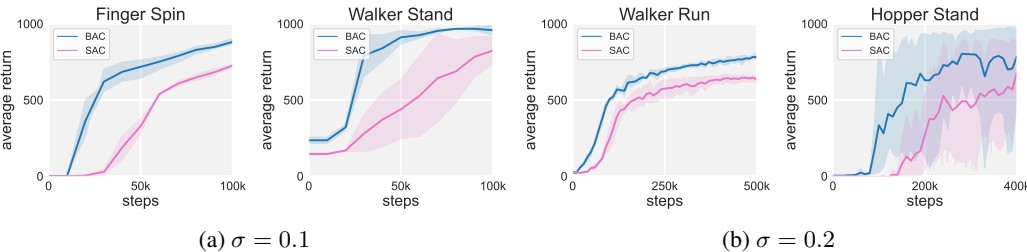

Figure 28: Results in noisy environments: (a) in noisy FingerSpin and WalkerStand tasks with $\sigma = 0.1$; (b) in noisy WalkerRun and HopperStand tasks with a server noise $\sigma = 0.2$.

## F.3 ILLUSTRATIVE EXAMPLE ON THE FAILURE-PRONE SCENARIO.

We provide a typical failure-prone scenario to illustrate the effectiveness of our operator.

**Task description.** As shown in Figure 29, a small particle spawns in a 2D continuous space of $[0, 10] \times [0, 10]$. The particle could take any random moves inside the space with a length of 0.1. The objective is to let the particle hit the small hole of radius 0.1 at $(10, 5)$. In other words, the particle receives a non-zero reward if and only if it is in the hole. Starting from a random policy, the particle has to explore the space and find the hole.

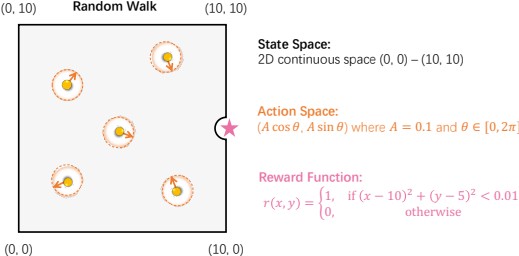

Figure 29: We construct a failure-prone scenario: Random Walk. The yellow particle has to explore the 2D space, and the target is to reach the small hole around $(10, 5)$ (pink star).

**$Q$-value comparison.** Only 10 of 100000 samples have reached the hole in the replay buffer. Figure 30 shows the $Q$-value heatmaps with the standard Bellman operator and our proposed BEE operator after 100, 200, and 500 $Q$-learning iterations. $Q$-values learned by BEE operator are much closer to the expected ones in limited iterations.

Let's dive deeper. Given $\bar{s}$ is one of the successor of a tuple $(s, a)$, the target update $r + \gamma \mathbb{E}_{a' \sim \pi} Q(s', a')$ for Standard Bellman Operator, only focuses on actions $a'$ from the current policy $\pi$, ignoring a more optimal one $\bar{a}'$. Thus $Q(s, a)$ which should be valued higher is underestimated. Then next policy derived from current misleading $Q$ may perfer not to sample $(s, a)$ again as it does not have a high value. Thus, algorithms based on the standard Bellman operator might take a substantially longer time to re-encounter these serendipities for training with decreased sample efficiency. In contrast, the BEE operator extracts the best actions in the replay buffer to construct referenced value to the $Q$-value function thus mitigates such underestimation issue.

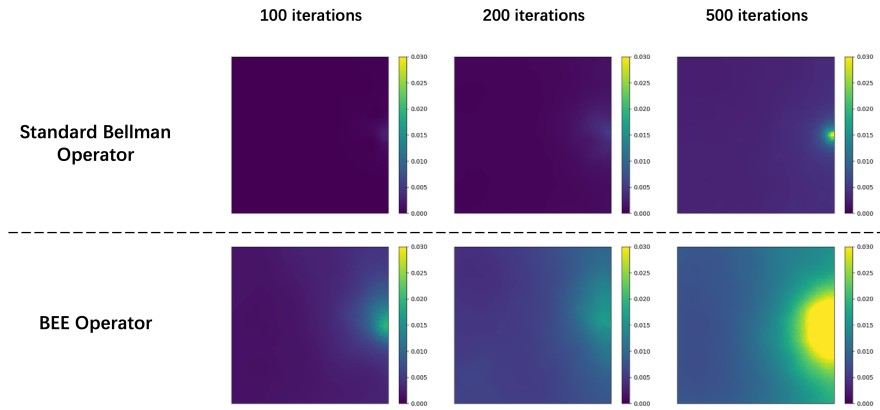

Figure 30: $Q$-value heatmaps with standard Bellman operator and the BEE operator after 100, 200 and 500 iterations.

### F.4 EFFECTIVENESS IN SPARSE-REWARD TASKS

We conduct experiments in sparse reward tasks to further demonstrate the generalizability of BEE operator. We evaluate in both robot locomotion and manipulation tasks, based on the sparse reward version of benchmark tasks from Meta-World (Yu et al., 2019), panda-gym (Gallouédec et al., 2021), ROBEL (Ahn et al., 2020). Here is the task description:

Meta-World manipultation tasks are based on a Sawyer robot with end-effector displacement control.

- coffee button: Push a button on the coffee machine whose position is randomized.
- hand insert: Insert the gripper into a hole.
- door open: Open a door with a revolving joint. Randomize door positions.

Panda-gym manipultation tasks are based on a Franka Emika Panda robot with joint angles control.

- PandaReachJoints: A target position must be reached with the gripper. This target position is randomly generated in a volume of $30cm \times 30cm \times 30cm$.

ROBEL quadruped locomotion tasks are based on a D'Kitty robot with 12 joint positions control.

- DKittyStandRandom: The D'Kitty robot needs to reach a pose while being upright from a random initial configuration. A successful strategy requires maintaining the stability of the torso via the ground reaction forces.
- DKittyOrientRandom: The D'Kitty robot needs to change its orientation from an initial facing direction to a random target orientation. A successful strategy requires maneuvering the torso via the ground reaction forces while maintaining balance.

As shown in Figure 31, our BAC surpasses the baselines by a large margin.

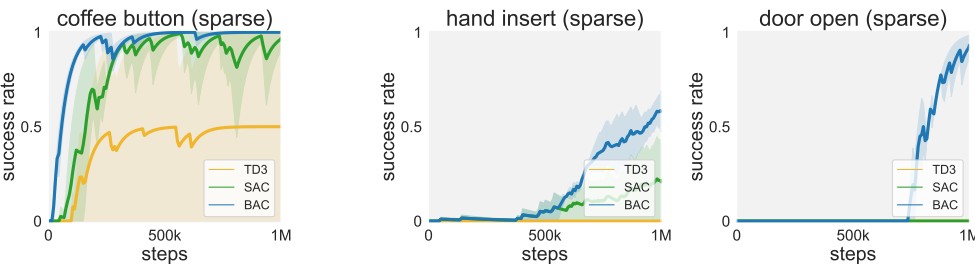

(a) `Meta-World` manipulation tasks on end-effector displacement control, using a Sawyer robot.

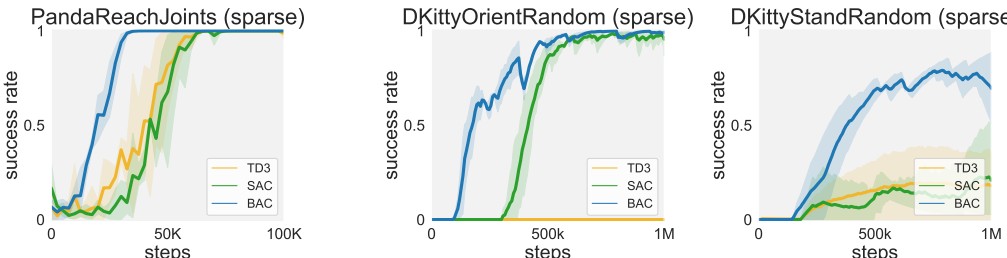

(b) `Panda-gym` manipulation tasks on joint angles control, using a Franka Emika Panda robot.

(c) `ROBEL` quadruped locomotion tasks, on 12 joint positions control, using a ROBEL D'Kitty robot.

Figure 31: **Sparse reward tasks.** BAC outperforms the baselines on six sparse reward tasks across various control types and robotic platforms, including manipulation and locomotion tasks.

### F.5 TASK VISUALIZATIONS IN FAILURE-PRONE SCENARIOS

**HumanoidStandup.** HumanoidStandup, provided by MuJoCo (Todorov et al., 2012), is a challenging locomotion task. The environment begins with the humanoid lying on the ground, and the goal is to enable the humanoid to stand up and then keep it standing by applying torques on the various hinges. The agent takes a 17-element vector for actions.

In the HumanoidStandup task, BAC demonstrates a significantly superior performance than all other algorithms. With average returns reaching approximately 280,000 at 2.5 million steps and 36,000 at 5 million steps, BAC surpasses other algorithms whose asymptotic performance peak at around 170,000, as illustrated in Figure 32.

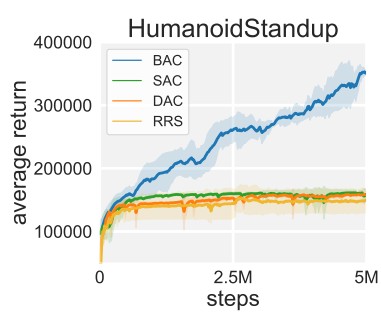

Figure 32: Learning curves of BAC and other baselines on HumanoidStandup task.

Visualization in Figure 33 depicts that the BAC agent can quickly achieve a stable standing pose. In contrast, SAC agent ends up in an unstable, swaying kneeling position, DAC ends up sitting on the ground, and the RRS agent, regrettably, is seen rolling around.

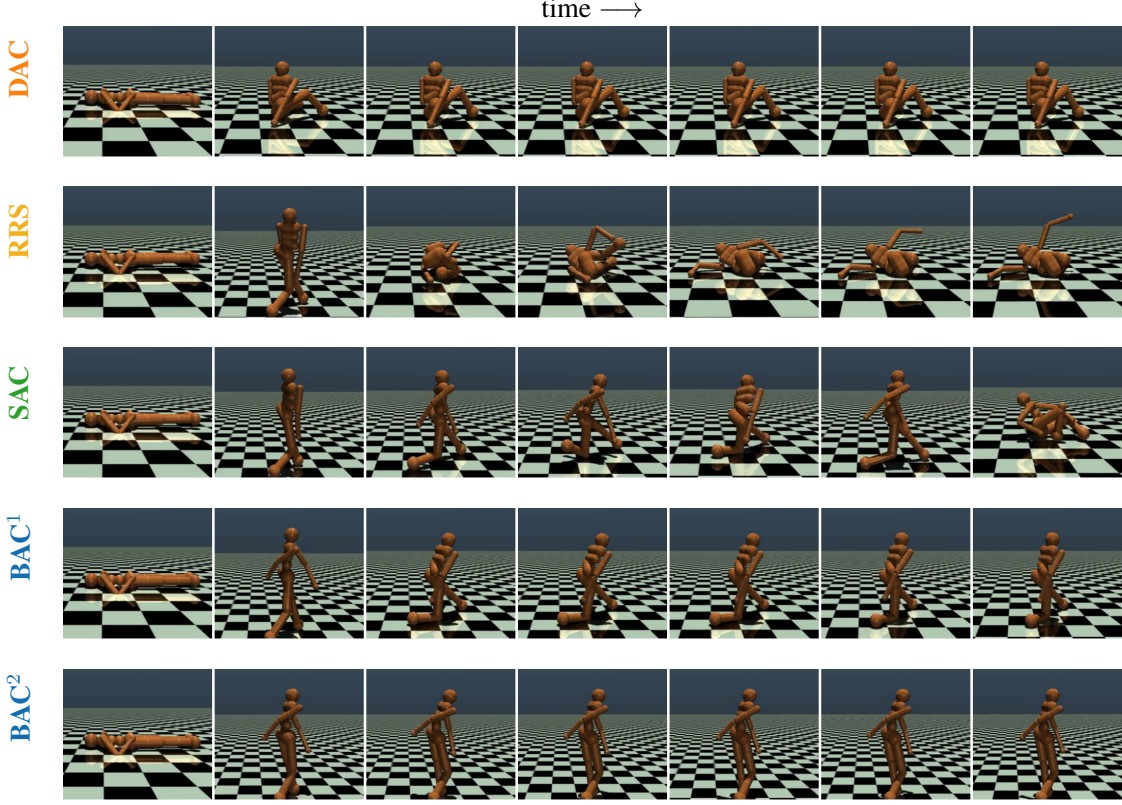

Figure 33: Visualization on HumanoidStandup task. BAC[1] is the visualization using the learned policy at 2.5M steps, and BAC[2] reveals the behaviors learned at 5M steps. For DAC, RRS and SAC, we visualize the learned policy at 5M steps.

**DogRun.** DogRun, provided by the DMControl (Tunyasuvunakool et al., 2020), is a challenging task with a high-dimensional action space ($\mathcal{A} \in \mathbb{R}^{38}$). The task is based on a sophisticated model of a Pharaoh Dog, including intricate kinematics, skinning weights, collision geometry, as well as muscle and tendon attachment points. This complexity makes the DogRun task extremely difficult for algorithms to learn and control effectively.

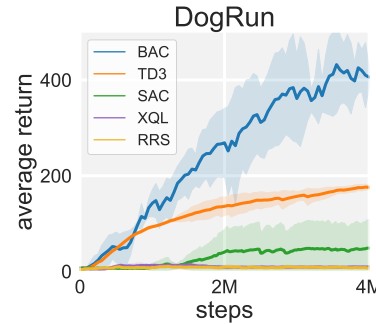

Figure 34: Learning curves BAC and other baselines on DogRun task.

We conducted extensive experiments in the DogRun task to compare the performance of BAC against other state-of-the-art algorithms. Here, we include Extreme Q-Learning (XQL) (Garg et al., 2023) as our baseline, which falls into the MaxEntropy RL framework but directly models the maximal $Q$ value. The results, depicted in Figure 34, reveal that BAC significantly surpasses its counterparts, attaining higher average returns in fewer interactions. It demonstrates a remarkable capability of learning to control the high-dimensional, complex robot, such as facilitating the dog's run. To the best of our knowledge, it is the first documented result of model-free methods effectively tackling the challenging DogRun task.

In addition to the quantitative results, we also offer a visualization of keyframes in the trajectory in Figure 35. Here, the superior performance of BAC becomes even more apparent. While competing algorithms struggle to prevent the dog from falling, BAC successfully achieves a running motion. This aptitude for handling complex, high-dimensional tasks further reaffirms the efficacy and robustness of BAC when dealing with failure-prone scenarios.

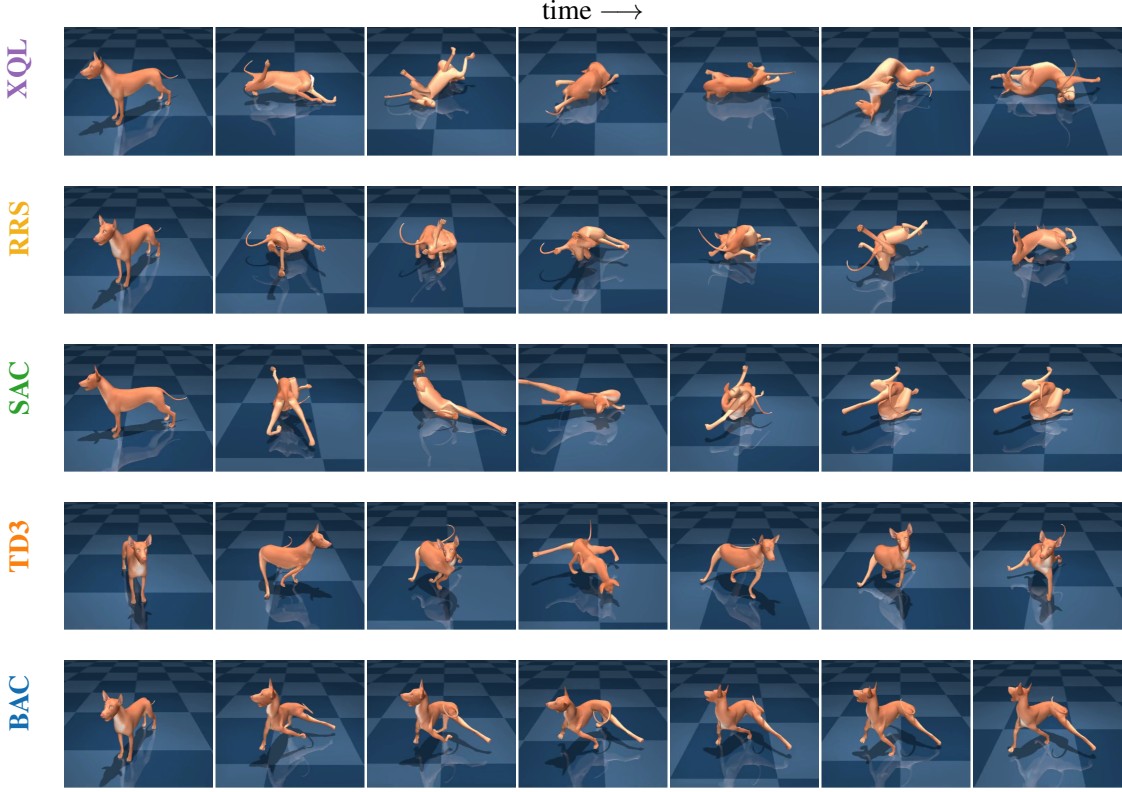

Figure 35: Visualization on DogRun task. We visualize the keyframes of the trajectories induced by the learned policy of each algorithm at 4M steps.

# G    MORE BENCHMARK RESULTS

Given that MuJoCo benchmark tasks have been solved well by popular baselines, we conduct experiments on the more complex locomotion and manipulation tasks from DMControl and Meta-World for further evaluation of BAC and the baselines. Currently, several tasks in DMControl and Meta-World pose a formidable challenge that stumps most model-free methods. Notably, BAC has demonstrated its effectiveness by successfully solving many of these challenging tasks.

## G.1    EVALUATION ON DMCONTROL BENCHMARK TASKS

We tested BAC and its variant, BEE-TD3, on 15 continuous control tasks from DMControl. BAC successfully solves many challenging tasks like HumanoidRun, DogWalk, and DogRun, where both SAC and TD3 fail. Also, BEE-TD3 boosts TD3's performance by a large margin, demonstrating the generalizability of the BEE operator.

**Performance comparison.**    Training curves for 15 DMControl tasks are shown in Figure 36. For simple locomotion/manipulation tasks (*e.g.*, HopperStand, WalkerStand, CupCatch), we generally find that while SAC's eventual performance is competitive with BAC, BAC shows better sample efficiency. In the more complex, failure-prone tasks (*e.g.*, HumanoidWalk, HumanoidRun, DogWalk, and DogRun), BAC significantly surpasses SAC. As shown in the visualizations[3], SAC struggles to learn meaningful behaviors in Dog Run, whereas the BAC agent yields superior performance.

**Additional Metrics.**    We report additional (aggregate) performance metrics of BAC and SAC on the set of 15 DMControl tasks using the `rliable` toolkit (Agarwal et al., 2021). As shown in Figure 37, BAC outperforms SAC in terms of Median, interquantile mean (IQM), Mean, and Optimality Gap.

**Hyperparameters for DMControl tasks.**    The corresponding hyperparameters are listed in Table 7.

Table 7: Hyperparameter settings for BAC in DMControl tasks.

|  | CartPole Swingup | Finger Spin | Cup Catch | Cheetah Run | Reacher Easy |
|---|---|---|---|---|---|
| $\lambda$ | 0.4 | 0.5 | 0.5 | 0.5 | 0.5 |
| $\tau$ | 0.7 | 0.7 | 0.7 | 0.7 | 0.7 |
|  | Reacher Hard | Hopper Stand | Walker Stand | Walker Run | Quadruped Walk |
| $\lambda$ | 0.5 | 0.5 | 0.5 | 0.5 | 0.5 |
| $\tau$ | 0.7 | 0.7 | 0.7 | 0.8 | 0.7 |
|  | Humanoid Walk | Humanoid Run | Acrobot Swingup | Dog Walk | Dog Run |
| $\lambda$ | 0.4 | 0.5 | 0.5 | 0.5 | 0.6 |
| $\tau$ | 0.7 | 0.7 | 0.7 | 0.7 | 0.7 |

---

[3]Please refer to https://beeauthors.github.io for videos or Section D.4 for key frames.

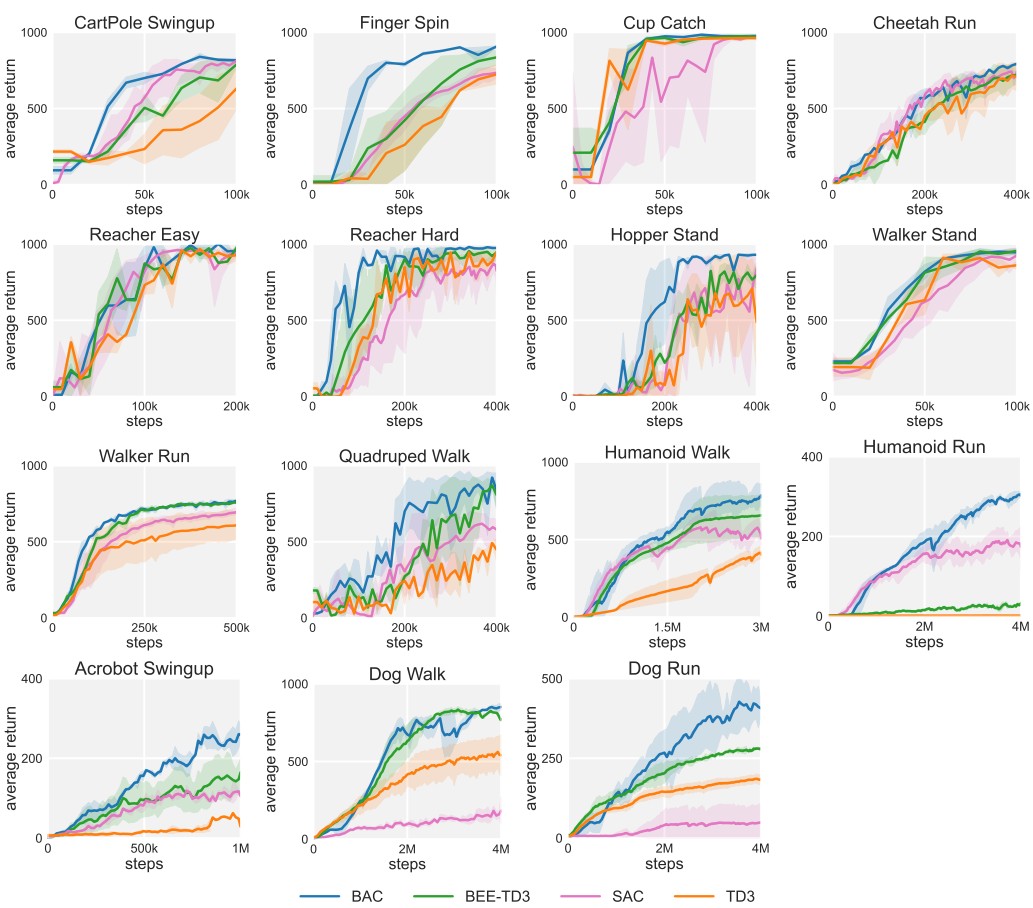

Figure 36: **DMControl tasks.** Training curves of BAC , BEE-TD3, SAC, TD3 in DMControl benchmark tasks. Solid curves depict the mean of four trials and shaded regions correspond to the one standard deviation.

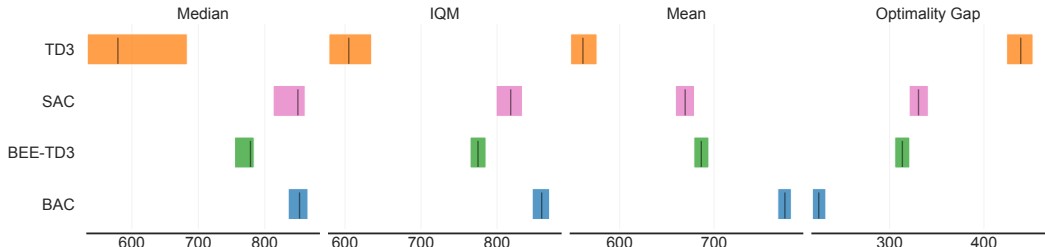

Figure 37: **Reliable metrics on DMControl tasks.** Median, IQM, Mean (higher values are better), and Optimality Gap (lower values are better) performance of BAC , BEE-TD3 and baselines (SAC, TD3) on the 15 DMControl tasks.

**Trajectory Visualizations.** Figure 38 provides visualizations of trajectories generated by BAC on five tasks from DMControl. For each trajectory, we display seven keyframes.

time ⟶

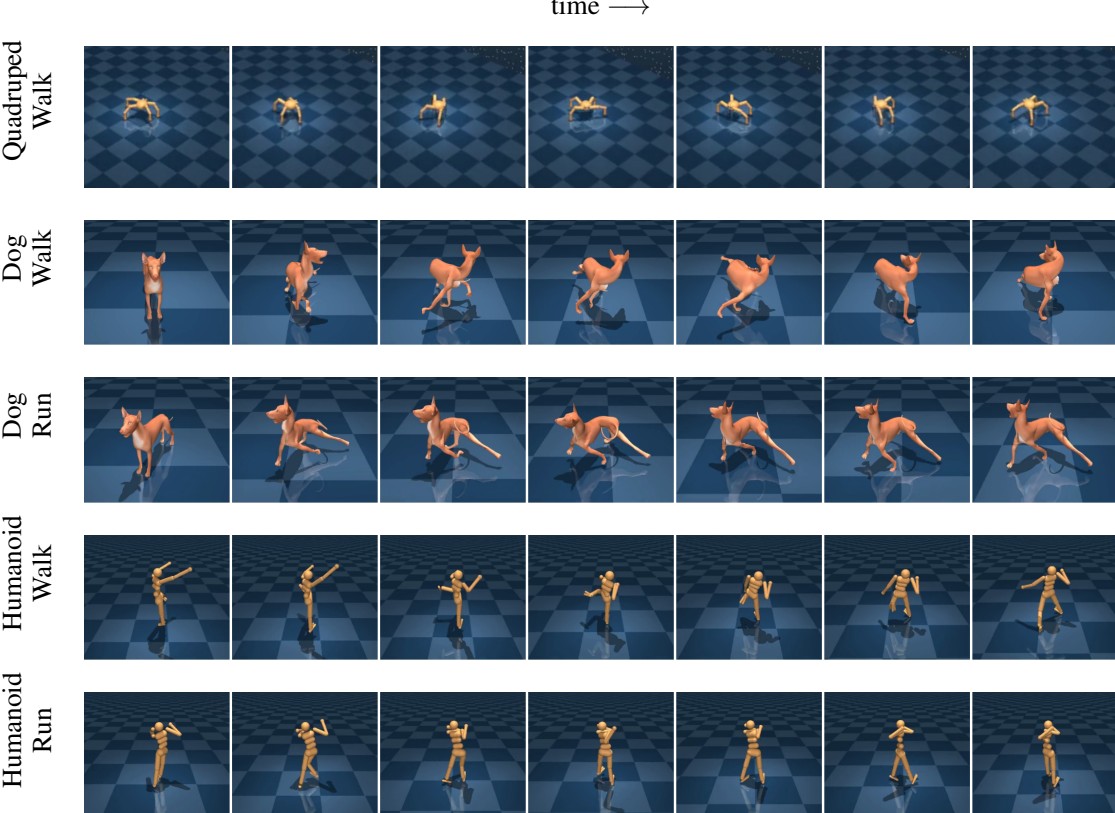

Figure 38: **Trajectory Visualizations.** Visualizations of the learned policy of BAC on five DMControl benchmark tasks.

## G.2   EVALUATION ON META-WORLD BENCHMARK TASKS

**Performance comparison.**   In Figure 39, we present learning curves of both the success rate and average return for twelve individual Meta-World tasks. Note that we conduct experiments on the goal-conditioned versions of the tasks from Meta-World-v2, which are considered harder than the single-goal variant.

In tasks typically categorized as simple, where both SAC and TD3 succeed within 1M steps, it is noteworthy that BAC still outperforms in terms of sample efficiency.

In tasks involving complex manipulation, such as pick place, basketball, hand insert, coffee push and hammer, BAC exhibits strong performance. Consider the hammer task, while SAC and TD3 occasionally achieve serendipitous successes before reaching 500K steps, their $Q$-value estimations are susceptible to misguidance by the inferior follow-up actions that occur frequently, resulting in a sustained low success rate. In contrast, BAC efficiently exploits the value of success and mitigates the impact of inferior samples on the $Q$-value, leading to a significant performance improvement beyond 500K steps, and finally surpasses SAC and TD3 by a large margin in terms of success rate.

These results highlight the promising potential of BAC in manipulation tasks.

**Trajectory Visualizations.**   Successful trajectories for one simple task and five aforementioned complex tasks are visualized in Figure 40. For each trajectory, we display seven keyframes.

**Hyperparameters for Meta-World tasks.**   The corresponding hyperparameters are listed in Table 8. We use a simple set of hyperparameters of $\lambda = 0.5$ and $\tau = 0.7$ for these twelve tasks, which already yields satisfactory performance, without the need for intensive hyperparameter-tuning efforts.

Table 8: Hyperparameter settings for BAC in Meta-World tasks.

|  | basketball | button press wall | coffee button | coffee push | door open | door unlock |
|---|---|---|---|---|---|---|
| $\lambda$ | 0.5 | 0.5 | 0.5 | 0.5 | 0.5 | 0.5 |
| $\tau$ | 0.7 | 0.7 | 0.7 | 0.7 | 0.7 | 0.7 |
|  | drawer open | hammer | hand insert | pick place | plate slide | window open |
| $\lambda$ | 0.5 | 0.5 | 0.5 | 0.5 | 0.5 | 0.5 |
| $\tau$ | 0.7 | 0.7 | 0.7 | 0.7 | 0.7 | 0.7 |

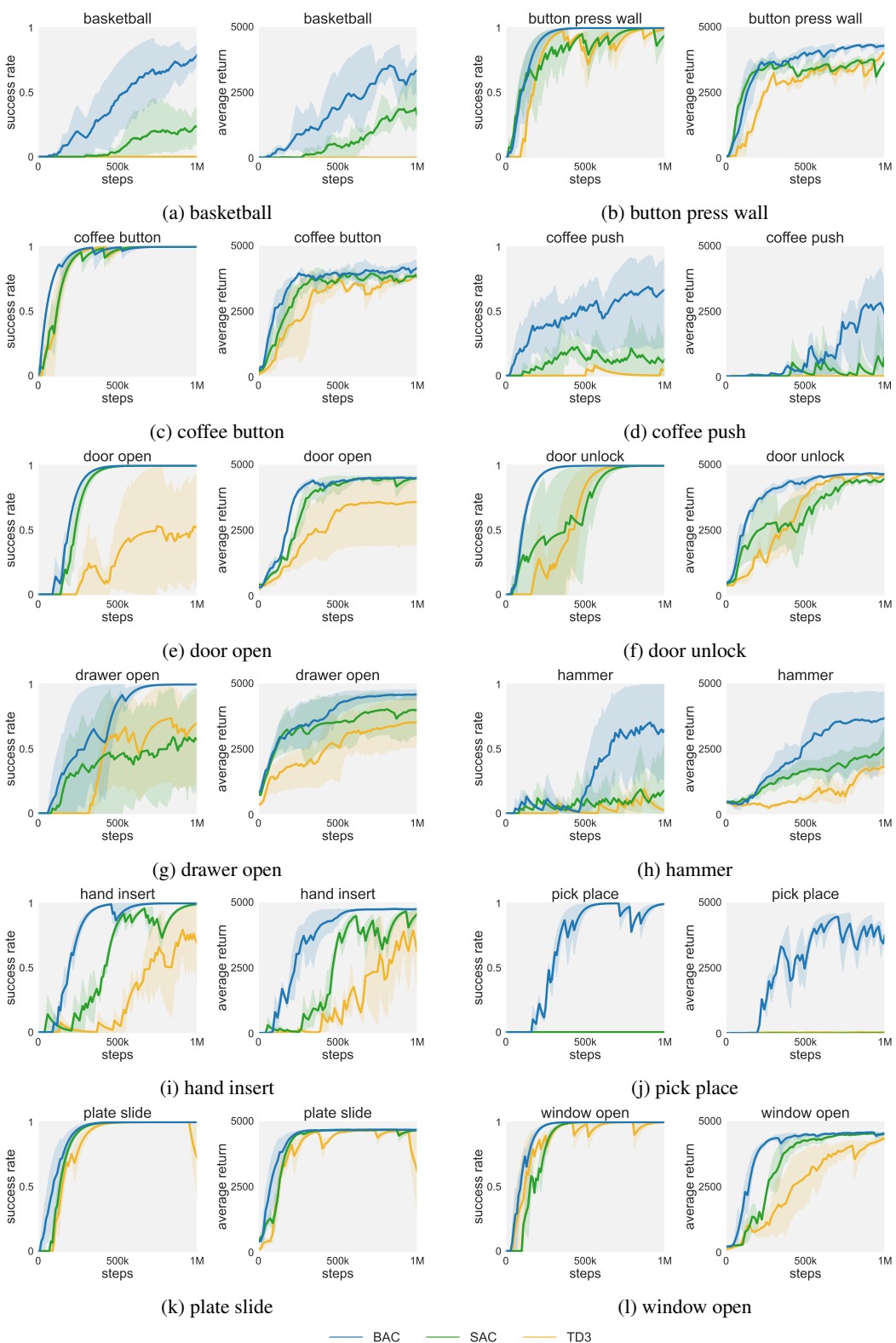

Figure 39: **Individual Meta-World tasks.** Success rate and average return of BAC , SAC, TD3 on twelve manipulation tasks from MetaWorld (sorted alphabetically). Solid curves depict the mean of four trials and shaded regions correspond to the one standard deviation.

time $\longrightarrow$

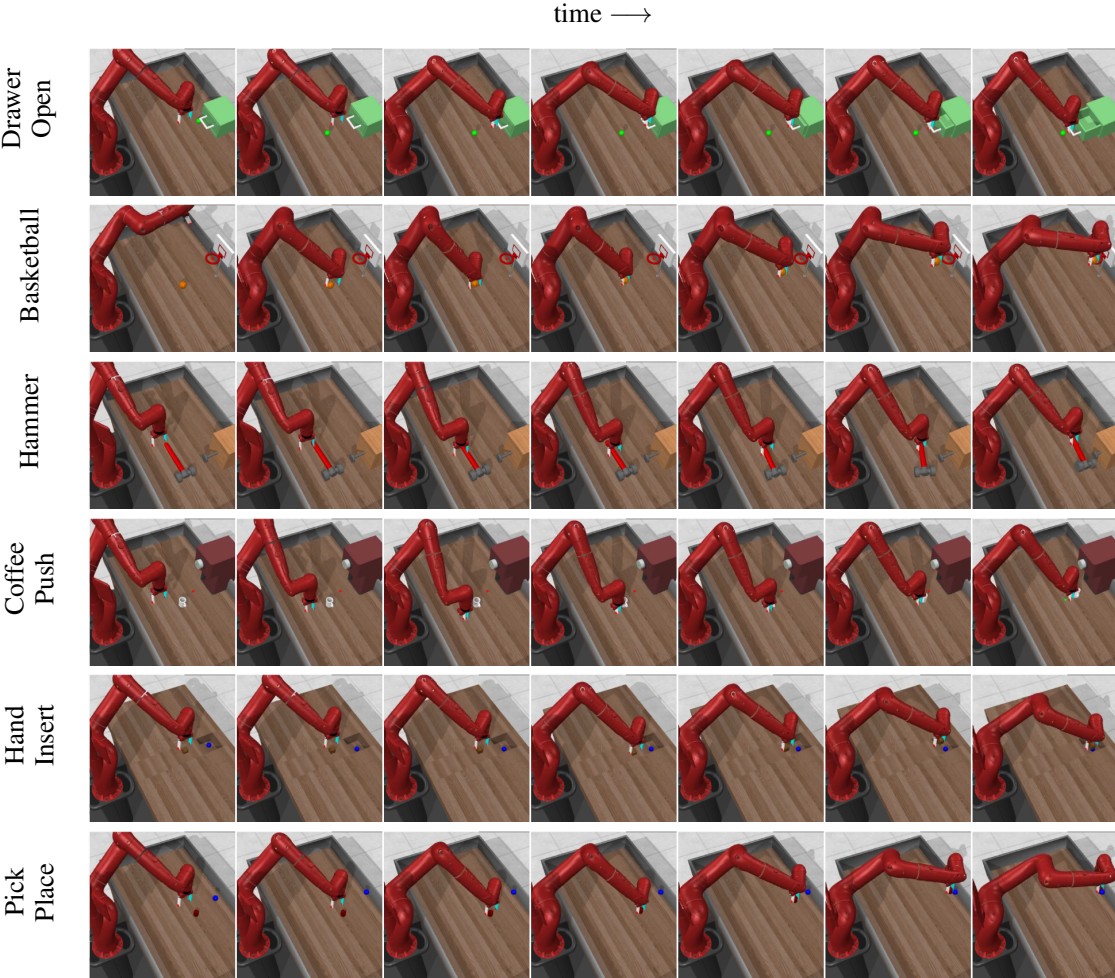

Figure 40: **Trajectory Visualizations.** We visualize trajectories generated by BAC on six Meta-World tasks. Our method (BAC) is capable of solving each of these tasks within 1M steps.

