# OpenReview forum: "Seizing Serendipity: Exploiting the Value of Past Success in Off-Policy Actor Critic"
_ICLR.cc/2024/Conference — Submitted to ICLR 2024_

### Official Review · Reviewer_7mFW · 2023-10-31

**Soundness:** 3 good
**Presentation:** 2 fair
**Contribution:** 2 fair
**Rating:** 6
**Confidence:** 3

**Summary:**

This work focuses on the Q-function value overestimation issue. Observing that the overestimation issue will latter becomes underestimation during the learning process. Thus motivated, this work proposes the Blended Exploitation and Exploration (BEE) operator to take advantage of the historical best-perforation actions. The proposed operator is then used in both model-free and model-based settings and show better performance than previous methods.

**Strengths:**

1. The proposed BEE operator utilizes the Bellman exploitation operator and exploration operator to address the under-exploitation issue. The proposed operator can be easily incorporated into the RL algorithms.
2. The experiments show that the proposed operator can effectively reduce the estimation error and achieve better performance comparing with other RL algorithms.

**Weaknesses:**

1. The terminology can be misleading. The overestimation issue in the Q-value approximation generally is due to the changing order of expectation and $\max$. It is incorrect to say that  the $Q$-function will have "underestimation when encountering successes" in Fig 1 (a). The authors need to clarify the context and difference of the statement in order to avoid confusion.
2. In order to investigate on the under-exploitation, the metric $\Delta(\cdot,\cdot)$ is defined on the current Q-function approximation. Intuitively,   $\Delta(\cdot,\cdot)$ shows that the current Q-function approximation can be either overestimate or underestimate given different policy, i.e., $\mu_k$ and $\pi_k$. It is unclear what is the meaning of this metric. Considering most of the algorithm will update the policy and Q-function approximation at the same time, e.g., Actor-Critic, the Q-function should be evaluated under the current policy instead of the policy obtained earlier. The authors need to clarify why the definition here makes sense for the under-exploitation investigation.

**Questions:**

See the weakness above.

---

> ### Author Response · Authors · 2023-11-16
> **Response to Reviewer 7mFW**
>
> Thank you for your valuable comments and suggestions, the detailed responses regarding each comment are listed below. If you have any other questions, please post them and we are happy to have further discussions.
>
> > **Q1: The terminology can be misleading. The overestimation issue in the $Q$-value approximation generally is due to the changing order of expectation and max. It is incorrect to say that the $Q$-function will have "underestimation when encountering successes" in Fig 1 (a). The authors need to clarify the context and difference of the statement in order to avoid confusion.**
>
> Thanks for your suggestions. We have refined it in our revised paper.
>
> * We removed the terminology in Fig 1(a).
> * We refined the caption of Fig 1(a) as, "When current policy generated action is inferior to the best action in the replay buffer, which usually occurs in the later stage of training (referred to as the under-exploitation stage), SAC is more prone to underestimation pitfalls than BAC."
>
>
>
>
> > **Q2: In order to investigate on the under-exploitation, the metric Δ(⋅,⋅) is defined on the current Q-function approximation. Intuitively, Δ(⋅,⋅) shows that the current Q-function approximation can be either overestimate or underestimate given different policy, i.e., $\mu_k$ and $\pi_k$. It is unclear what is the meaning of this metric. Considering most of the algorithm will update the policy and Q-function approximation at the same time, e.g., Actor-Critic, the Q-function should be evaluated under the current policy instead of the policy obtained earlier. The authors need to clarify why the definition here makes sense for the under-exploitation investigation.**
>
>
>
> Sorry for any confusion in our initial presentation. We've revised the notation for clarity:
>
>
> $\Delta(\mu_k, \pi_k) = \mathbb{E}_s[\max\_{a\sim \mu_k}Q^{\mu_k}(s,a) - \mathbb{E}\_{a\sim \pi_k} [Q^{\pi_k}(s,a)-\omega(s,a\vert \pi_k)]]
> $.
>
> This formula shows the expected difference between two different types of calculated Q-values. Specifically:
>
> * $\max_{a\sim \mu_k}Q^{\mu_k}(s,a)$:  This is the maximum Q-value according to the policy mixture $\mu_k$ learned using offline RL techniques from the replay buffer. It approximates the value of the best policy that could be derived from the replay buffer.
> * $\mathbb{E}_{a\sim \pi_k} Q^{\pi_k}(s,a)$: This corresponds to the expected Q-value under the current policy $\pi_k$. And the exploration bonus $-\omega(s,a\vert \pi_k)$ could also be considered when computing.
>
>
>
>
> Thus, $\Delta(\mu_k, \pi_k)$ quantifies the gap between the optimal actions in the replay buffer and those generated by the current policy $\pi$. A positive $\Delta(\mu_k, \pi_k)$ indicates the existence of better actions in the replay buffer that have not been fully exploited.
>
>
>
> Finally, we hope we resolve all of your concerns. We have refined our explanations in the rebuttal revision according to your suggestions. Thanks again for your suggestions!

---

> > ### Comment · Reviewer_7mFW · 2023-11-23
> > **Ack author's response**
> >
> > I thank the author's response on my questions. I will keep my original score.

---

> > > ### Author Response · Authors · 2023-11-23
> > > **Thank you for your responsible reply!**
> > >
> > > Dear reviewer,
> > >
> > > We thank the reviewer for your valuable comments and suggestions, which provide helpful guidance to improve the quality of our paper! We really appreciate your efforts!
> > >
> > > Best wishes!
> > >
> > > The authors.

---

### Official Review · Reviewer_FAWm · 2023-10-31

**Soundness:** 3 good
**Presentation:** 3 good
**Contribution:** 3 good
**Rating:** 6
**Confidence:** 4

**Summary:**

This paper presents the Blended Exploitation and Exploration (BEE) operator, which addresses the issue of value underestimation during the exploitation phase in off-policy actor-critic methods. The paper highlights the importance of incorporating past successes to improve Q-value estimation and policy learning. The proposed BAC and MB-BAC algorithms outperform existing methods in various continuous control tasks and demonstrate strong performance in real-world robot tasks.

**Strengths:**

- The paper addresses an important issue in off-policy actor-critic methods and proposes a novel approach to improve Q-value estimation and policy learning.
- The BEE operator is simple yet effective and can be easily integrated into existing off-policy actor-critic frameworks.
- The experimental results demonstrate the superiority of the proposed algorithms in various continuous control tasks and real-world robot tasks.

**Weaknesses:**

1. The novelty of the proposed approach is limited.
2. The choice of $\lambda$ is largely empirical and requires extra manipulation in new tasks.
3. The paper only provides basic theoretical analysis, such as the accurate policy evaluation and the guarantee of policy improvement. The benefit of linearly combining two Q-value functions is not discussed theoretically.
4. The experiments are conducted in continuous control tasks with dense rewards. The exploration ability can be better evaluated in environments with sparse rewards.
5. There is a lack of discussions with related papers (See Question 3).

**Questions:**

1. Emprically, the BAC algoithm will only be more efficient in exploiting the replay buffer. The exploration still rely on the maximum-extropy formulation in SAC. Then why can BAC perform significantly better than SAC in failure-prone scenarios such as HumanoidStandup, as if BAC can better explore the unknown regions?
2. Can you discuss or exhibit the performance of BAC in some tasks with sparse rewards? This can demonstrate the generalizability of the proposed approach.
3. What are the advantages of BAC compared with prioritized replay methods [1,2] or advantage-based methods [3]? These methods are related to BAC in that they also exploit the replay buffer with inductive bias, so they should be mentioned in the paper.

[1] Sinha, S., Song, J., Garg, A. &amp; Ermon, S.. (2022). Experience Replay with Likelihood-free Importance Weights. Proceedings of The 4th Annual Learning for Dynamics and Control Conference.

[2] Liu, X. H., Xue, Z., Pang, J., Jiang, S., Xu, F., & Yu, Y. (2021). Regret minimization experience replay in off-policy reinforcement learning. Advances in Neural Information Processing Systems, 34, 17604-17615.

[3] Nair, A., Gupta, A., Dalal, M., & Levine, S. (2020). Awac: Accelerating online reinforcement learning with offline datasets. arXiv preprint arXiv:2006.09359.


I am willing to raise my score if my concerns for weaknesses and questions are adequately discussed.

---

> ### Author Response · Authors · 2023-11-16
> **Response to Reviewer FAWm (1/2)**
>
> Thank you for your comments and suggestions! We provide clarification to your questions and concerns as below.  If you have any additional questions or comments, please post them and we would be happy to have further discussions.
>
> > **W1: The novelty of the proposed approach is limited.**
>
> Our novelty could be summarized below:
>
> * While existing off-policy RL algorithms primarily focused on alleviating value over-estimation, our work reveals the long-neglected underestimation phenomenon in the latter stage of the online RL training process. We have provided extensive investigations into this issue (more in Appendix E), offering fresh insights into this overlooked area.
> * Our findings shed light on a new way to cure underestimation: incorporating sufficient exploitation by leveraging techniques from offline RL. Our proposed BEE operator is simple and versatile and achieved great performance in our empirical experiments.
> * Furthermore, our study **demystifies the misperception** that offline RL is over-conservative and incompatible with the online RL setting. It suggests a new paradigm that incorporates exploitation ingredients from offline RL to enhance pure online RL.
>
> > **W2: The choice of $\lambda$ is largely empirical and requires extra manipulation in new tasks.**
>
> There might be a misunderstanding here. Simply using $\lambda=0.5$ is sufficient for strong performance across all the 50 tasks in our experiments.  We provide clarification on the hyperparameter robustness; please refer to our **General Response**.
>
> > **W3: The paper only provides basic theoretical analysis, such as the accurate policy evaluation and the guarantee of policy improvement. The benefit of linearly combining two Q-value functions is not discussed theoretically.**
>
> The focus of our study is to provide a strong practical algorithm rather than presenting a theoretical study to provide lots of performance bounds. Actually, in order to provide a versatile framework that is capable of leveraging wide choices of exploration terms $\omega(\cdot|\pi)$ and Bellman exploitation operator evaluation methods (see Section 4.4 for details), it becomes pretty challenging for anyone to provide further theoretical analysis. Since different instantiation choices will introduce different theoretical properties. However, as mentioned by the reviewer, we have empirically verified the effectiveness via extensive experiments in both the simulated environments and a real robot. We hope this can justify the strengths of our algorithm.
>
> > **Q1: Emprically, the BAC algoithm will only be more efficient in exploiting the replay buffer. The exploration still rely on the maximum-extropy formulation in SAC. Then why can BAC perform significantly better than SAC in failure-prone scenarios such as HumanoidStandup, as if BAC can better explore the unknown regions?**
>
> - First, we'd like to clarify that our exploration term $\omega(\cdot|\pi)$ is not restricted to the entropy term as in SAC; other exploration designs are also applicable, as discussed in Section 4.4. We use the entropy term as in SAC mainly due to its simplicity.
> - Second, as we have discussed in our paper, SAC is not very effective to exploit the high-quality samples in the replay buffer. This can have a great impact on failure-prone tasks, as informative, successful samples can be scarce in policy-generated samples. Blindly sampling batches of data from the replay buffer can be very ineffective in extracting useful information for value updates. For example, in the HumanoidStandup task,  SAC can only maintain short-standing postures before failing, as seen from the trajectory visualization (Figure 33 or Video in https://beeauthors.github.io/ ). This indicates that while SAC has explored high-value transitions, it struggles to exploit them effectively. In this case, fully exploiting the high-quality samples in the replay buffer for value updates is essential to achieve efficient policy learning.
> - We present more illustrative results to showcase the effectiveness of our method in failure-prone scenarios in our revision; please refer to Appendix F.3.

---

> ### Author Response · Authors · 2023-11-16
> **Response to Reviewer FAWm (2/2)**
>
> > **Q2: Can you discuss or exhibit the performance of BAC in some tasks with sparse rewards? This can demonstrate the generalizability of the proposed approach. The exploration ability can be better evaluated in environments with sparse rewards.**
>
> Thanks for your constructive comment!
> * We have conducted experiments on six sparse reward tasks across various control types and robotic platforms, including manipulation and locomotion tasks. And BAC outperforms the baselines a lot. Please refer to Figure 31 in our revised paper.
> - Besides, the illustrative example in Appendix F.3 is also a sparse reward setting, showcasing the BEE's effectiveness.
>
> > **Q3: What are the advantages of BAC compared with prioritized replay methods [1,2] or advantage-based methods [3]? These methods are related to BAC in that they also exploit the replay buffer with inductive bias, so they should be mentioned in the paper.**
>
> Thank you for pointing out these interesting works.
> * **Prioritized replay methods**:
>     * We have included a discussion on prioritized replay methods in our revision and cited them accordingly. We'd like to emphasize that although prioritized replay methods follow a similar intuition of leveraging better samples from the replay buffer for enhanced value learning, BAC provides a much simpler and more lightweight manner. There is also no need to change the data pipeline as in prioritized replay methods.
>     * Second, we'd like to emphasize that BAC is compatible with prioritized replay methods. Recall the Bellman error used in policy evaluation, $\mathbb{E}_d[w(s,a)(Q(s,a)-\mathcal{B}Q(s,a))^2]$.  Here, the BEE operator designs a hybrid operator $\mathcal{B}^{\mu,\pi}$ focusing on blending exploitation and exploration, and prioritized replay methods concentrate on designing effective weights $w(s, a)$.
>
> * **Advantage-based methods**:  This class of methods has a different setting from ours. As we discussed in Related Works, AWAC is under a two-stage paradigm, performing offline RL on offline datasets and then followed by online fine-tuning. BAC is a **pure online RL** algorithm.
>
>
> Thanks again for your constructive suggestions. We hope the above can resolve your concerns and are glad to discuss further.

---

> > ### Comment · Reviewer_FAWm · 2023-11-21
> > **Re**
> >
> > Dear Authors,
> >
> > Thanks for the detailed rebuttal, which solves my concerns. I have raised my score accordingly.

---

> > > ### Author Response · Authors · 2023-11-21
> > > **Thank you for your inspiring reply!**
> > >
> > > Dear reviewer,
> > >
> > > Thank you for helping us improve the paper and for updating the score! We really appreciate your constructive comments and suggestions!
> > >
> > > Best wishes!
> > >
> > > The authors.

---

> ### Comment · Area_Chair_iF9f · 2023-11-20
>
> Dear FAWm,
>
> The author reviewer discussion period is ending soon this Wed. Does the author response clear your concerns w.r.t., e.g., limited novelty and hyperparamters? Are there still outstanding items that you would like to discuss more with the authors.
>
> Thanks again for your service to the community.
>
> Best,
> AC

---

### Official Review · Reviewer_kjkr · 2023-11-01

**Soundness:** 3 good
**Presentation:** 4 excellent
**Contribution:** 2 fair
**Rating:** 6
**Confidence:** 5

**Summary:**

Motivated by the problem of underestimating values in the training of SAC, this paper introduces the Blended Exploitation and Exploration (BEE) operator, which calculates the TD target based on a combination of the standard TD target and a high expectile of the return distribution. The authors integrate this operator in both model-free and model-based scenarios, followed by a comprehensive experimental evaluation.

**Strengths:**

1. The paper contains extensive experiment results on both simulation and real-world environments.
2. The paper is written clearly and easy to follow. Figure 1 provides a decent visualization of the underestimation issue.

**Weaknesses:**

1. The BAC method tunes its $\lambda$ and $\tau$ differently for tasks in MuJoCo and DMC (Table 1 & 5). It's questionable to claim superiority over other state-of-the-art (SOTA) methods like SAC and TD3, which use consistent hyperparameters (HP) across tasks. Adjusting HP for each task can inflate results as seen in Figure 5, which can be misleading. Why not showcase the automatic $\lambda$ tuning methods from Appendix B.3.3 in the main text if they're effective?

2. Figure 23 reveals that SAC, without the double-Q-trick, still underestimates in the Humanoid task. It's unclear if this is universally true. More convincing results would come from testing this across multiple tasks and providing absolute Q value estimates. I still suspect that Q underestimation largely stems from the double Q techniques, as suggested by the RL community [1]. For instance, OAC [1] introduces $\beta_{\text{LB}}$ to manage value estimation issues.

3. Presuming the Q value underestimation problem is widely recognized (which I invite the authors to contest), the paper seems to lack innovation. The BEE operator, at its core, appears to be a fusion of existing Bellman operators.

4. The statement "BEE exhibits no extra overestimation" seems conditional on specific $\lambda$ and $\tau$ values. For instance, using $\lambda = 1$ and $\tau = 1$ could induce overestimation.

[1] Ciosek, Kamil, et al. "Better exploration with optimistic actor critic." Advances in Neural Information Processing Systems 32 (2019).

**Questions:**

See Weakness

---

> ### Author Response · Authors · 2023-11-16
> **Response to Reviewer kjkr (Major Concern)**
>
> **Response summary.**  We thank the reviewer for their comments and feedback.  Based on the comments, it seems that the reviewer has one major concern and several other minor concerns. We provide clarifications below and would be happy to discuss further.
>
> > **Major Concern: Source of underestimation**
>
> We notice that the reviewer's major concern is that "underestimation largely stems from the double Q trick, and solving the widely recognized underestimation issue lacks novelty". We'd like to argue that this is not the whole story, and the detailed discussions are as follows:
> - **Double-Q-trick is not the only culprit:**  While the double-Q trick is thought as a possible source of underestimation [1], our findings suggest it's not the only factor.  Figure 24 shows that underestimation occurs in SAC and TD3 across 8 tasks without the double-Q trick.
> - **The optimization procedure of the actor-critic (AC) framework can also contribute to underestimation:**
>     - To recap, ideally, the Bellman update needs to solve $Q(s,a)\leftarrow r(s,a)+\gamma \mathbb{E}_s[\max_a Q(s,a)]$. However, as $\max\_a Q(s,a)$ operations are often impractical to calculate, so in AC framework, we typically iteratively evaluate the target Q-value as $\mathbb{E}\_{\pi}[Q(s,a)]$, while implicitly conducting the max-Q operation in a separate policy improvement step to learn policy $\pi$.
>     - Note that the ideal $\pi=\arg\max_{a\sim \pi} Q(s,a)$ is not possible to achieve practically within only a few policy gradient updates [2], hence the actual target value used in AC Bellman update $\mathbb{E}_{s,a\sim \pi}Q(s,a)$ can have a high chance to be smaller than $\mathbb{E}_s[\max_a Q(s,a)]$, causing underestimation. In other words, the non-optimal current policy in AC framework can also contribute to underestimation.
>
> * **Underestimation in the latter training stage:** In our paper, we have empirically shown that during the course of training, the replay buffer could contain more optimal actions as compared to the ones generated by the current policy. This becomes prevalent, especially in the latter training stage, when the replay buffer is filled with more high-performance samples. This leads to a notable shortfall of existing methods in exploiting the good samples in the replay buffer. Please refer to Appendix E.2 to E.4 for detailed analyses and empirical evidence.
> * **Incorporating sufficient exploitation is a natural cure:** In our work, we provide a simple solution (BEE operator) to fully exploit the good samples in the replay buffer to mitigate the underestimation while also greatly enhancing sample efficiency.

---

> ### Author Response · Authors · 2023-11-16
> **Response to Reviewer kjkr (Listed Questions)**
>
> > **Q1: The BAC method tunes its $\lambda$ and $\tau$ differently for tasks in MuJoCo and DMC (Table 1 & 5). It's questionable to claim superiority over other state-of-the-art (SOTA) methods like SAC and TD3, which use consistent hyperparameters (HP) across tasks. Adjusting HP for each task can inflate results as seen in Figure 5, which can be misleading.**
>
> We believe there is a misunderstanding about the HP tuning of our work.
> * As discussed in our **General Response**, simply using a single set of HP in our method can achieve SOTA performance throughout all 50 environments.
> * We did tune the HPs of SAC and TD3 baselines across benchmarks/tasks for improved performance, following suggestions from the original paper [3,4] (for MuJoCo), TD-MPC [5] paper (for DMC), Meta-World [6] paper (for MetaWorld), as detailed in Appendix D.  For clarity, we attach the HPs table for SAC and TD3 into our revision, please refer to Table 5 and Table 6.
>
> > **Q2: Why not showcase the automatic $\lambda$ tuning methods from Appendix B.3.3 in the main text if they're effective?**
>
> - We find that fixed $\lambda=0.5$ is already sufficient for strong performance, validated by extensive experiments across 50 tasks.
> - A simple yet efficient algorithm has been long advocated. Hence, we present BAC with minimal designs in the main text.  The automatic $\lambda$ tuning mechanism in the appendix is intended for interested readers to explore potential future refinements.
>
>
> > **Q3: Figure 23 reveals that SAC, without the double-Q-trick, still underestimates in the Humanoid task. It's unclear if this is universally true. More convincing results would come from testing this across multiple tasks and providing absolute Q value estimates. I still suspect that Q underestimation largely stems from the double Q techniques, as suggested by the RL community [1]. For instance, OAC introduces $\beta_{LB}$ to manage value estimation issues.**
>
> - Please refer to our previous response to the **major concern**.
> - Thanks for pointing out OAC.  But $\beta_{LB}$ in OAC could be perceived as a milder form of double-Q-trick and thus could not fully solve the underestimation issue.  Also, OAC needs efforts to tune three hyperparameters, as illustrated in its paper. We have added the corresponding discussion in the Related Work in our revision.
>
>
>
>
>
> > **Q4: Presuming the Q value underestimation problem is widely recognized (which I invite the authors to contest), the paper seems to lack innovation. The BEE operator, at its core, appears to be a fusion of existing Bellman operators.**
>
> We would like to emphasize that our method is not simply fusing existing Bellman operators. We revealed a long-neglected underestimation issue in an existing method, which could potentially hurt performance and sample efficiency (elaborated in our response to the **major concern**). To address this, we provide a natural and very simple solution to enable fully exploitation of high-quality samples in the replay buffer. More importantly, our work demystifies the misperception that offline RL is over-conservative and incompatible with the online RL setting, and also provides a new paradigm to incorporate exploitation ingredients from offline RL to enhance the performance of online RL.
>
>
>
>
>
> > **Q5: The statement "BEE exhibits no extra overestimation" seems conditional on specific $\lambda$ and $\tau$ values. For instance, using $\lambda$=1 and $\tau$=1 could induce overestimation.**
>
> * For $\lambda=1$: this case actually reduces to an in-sample learning offline RL algorithm (e.g., IQL in our instantiation), which is inherently conservative. We show in Appendix E.6 that this case does not exhibit overestimation. Besides, note that our BEE operator is defined on $\lambda\in (0,1)$, as stated in Definition 4.1.
> * For $\tau=1$:  BEE operator is not bounded to hyperparameter $\tau$. As detailed in Appendix B.3.1, implementing BEE with other in-sample learning offline techniques, such as SQL and EQL instead of IQL would not have $\tau$ at all.  We chose IQL primarily for its simplicity, and the original IQL paper also restricts $\tau < 1$.
>
>
> Thanks again for your comments. Please let us know if there are any remaining questions or concerns. We hope the reviewer can reassess our work in light of these clarifications and additional empirical results.

---

> ### Author Response · Authors · 2023-11-16
>
> **References**
>
> [1] Hasselt, Hado. "Double Q-learning." Advances in neural information processing systems, 2010, 23.
>
> [2] Chan A, Silva H, Lim S, et al. Greedification operators for policy optimization: Investigating forward and reverse kl divergences[J]. The Journal of Machine Learning Research, 2022, 23(1): 11474-11552.
>
> [3] Haarnoja T, et al. Soft actor-critic algorithms and applications[J]. arXiv preprint arXiv:1812.05905, 2018.
>
> [4] Fujimoto S, Hoof H, Meger D. Addressing function approximation error in actor-critic methods[C]//International conference on machine learning. PMLR, 2018: 1587-1596.
>
> [5] Hansen N, et al. Temporal Difference Learning for Model Predictive Control[C]. ICML, 2022.
>
> [6] Yu T, Quillen D, He Z, et al. Meta-world: A benchmark and evaluation for multi-task and meta reinforcement learning[C]//Conference on robot learning. PMLR, 2020: 1094-1100.

---

> > ### Comment · Area_Chair_iF9f · 2023-11-20
> > **author reviewer discussion is ending soon**
> >
> > Dear kjkr,
> >
> > The author reviewer discussion period is ending soon this Wed. Does the author response clear your concerns w.r.t., e.g., hyper parameters or there are still outstanding items that you would like the authors to address?
> >
> > Best,
> > AC

---

> ### Author Response · Authors · 2023-11-22
> **We sincerely look forward to your reply.**
>
> Dear reviewer kjkr,
>
> We first thank the reviewer again for the comments and suggestions. In our earlier response and revised manuscript, we have conducted additional experiments and provided detailed clarifications based on your questions and concerns.
>
> As we are ending the stage of the author-reviewer discussion soon, we kindly ask you to review our revised paper and our response and reconsider the score if our response has addressed your concerns.
>
> If you have any other questions, we are also pleased to respond. We sincerely look forward to your response.
>
> Best wishes!
>
> The authors

---

> > ### Comment · Reviewer_kjkr · 2023-11-22
> > **Thank you for your response!**
> >
> > The results using unified hyperparameters in Figure 17 and Figure 18 addressed my major concerns. I thus raised my score to 6.

---

> > > ### Author Response · Authors · 2023-11-22
> > > **Thank you for your inspiring reply!**
> > >
> > > Dear reviewer,
> > >
> > > Thank you for helping us improve the paper and for updating the score! We really appreciate your comments and suggestions!
> > >
> > > Best wishes!
> > >
> > > The authors.

---

### Author Response · Authors · 2023-11-16
**General Response to All Reviewers**

### **Response to concerns on hyperparameters**

We'd like to clarify that a single set of hyperparameters ($\lambda=0.5, \tau=0.7$)
is sufficient for our algorithm to achieve strong performance throughout all 50 experiments across MuJoCo, DMControl, Meta-World, Robel, and panda-gym benchmarks.
* In our experiments, for 41 out of 50 experiments,  we simply set $\lambda=0.5, \tau=0.7$ and consistently achieved SOTA performance as reported in our initial paper. The remaining 9 tasks are only slightly tuned, as we find that BAC can achieve even better performance.
* To address the reviewers' concerns, we include the results using the same $\lambda=0.5, \tau=0.7$ on the remaining 9 tasks. The results are presented in Figure 17 and Figure 18 of our revised paper, which demonstrate that BAC still outperforms all the baselines by a large margin.





---

### **Revision Summary**

We thank all the reviewers for their valuable and constructive comments.We have revised the paper as the reviewers suggested (highlighted in blue) and have addressed their comments. The summary of changes in the updated version of the paper is as follows:

1. (for Reviewer kjkr, FAWm) We added the results using unified hyperparameters in Figure 17 and Figure 18, Appendix B.4.
2. (for Reviewer kjkr) We attached the hyperparameters table of SAC and TD3 for different benchmarks in Table 5 and Table 6, Appendix D.
3. (for Reviewer kjkr) We added more visualizations on the underestimation issue of SAC/TD3 without double-Q-trick in Figure 24 and revised our analyses in Appendix E.4 for clarity.
5. (for Reviewer kjkr) We added a discussion on OAC-style work in Related Works.
6. (for Reviewer FAWm) We provided an illustrative example of the effectiveness of the BEE operator in failure-prone scenarios; please refer to Appendix F.3.
7. (for Reviewer FAWm) We added experiments on various sparse-reward tasks in Figure 31, Appendix F.4.
8. (for Reviewer FAWm) We added more discussion on prioritized replay methods in Related Works.
9. (for Reviewer 7mFW) We revised the terminology and caption in Figure 1(a).
10. (for Reviewer 7mFW) We revised the notation of $\Delta(\mu_k, \pi_k)$ for clarity.

---

### Meta-Review · Area_Chair_iF9f · 2023-12-05

**Metareview:**

The paper demonstrates and investigates the issue of under-estimation in late training stages of RL algorithms. A blended Bellman operator is proposed, consisting of both an exploitation one based on replay buffer data and an exploratory one based on entropy regularization. Dynamic programming properties of the blended Bellman operator are analyzed. Empirical study is conducted in both simulated and real-world tasks and performance improvements are seen using a single hyper parameter for trading off the two operators.

Strength: The demonstration of under exploitation appears novel and could possibly open up new research directions. The empirical study demonstrates the applicability of the proposed approach in real world robots.

Weakness: Most experiments are done with only 5 random seeds, making it hard to evaluate the statistical significance of the result. In particular, shaded regions are only an estimate of one standard derivation (which is hard to be accurate because again there are only 5 seeds) and the plots lack a measure of confidence. For example, none of the results in Figure 6 actually seem significant. Given the empirical nature of this work and the importance for having a single set of well-performing hyper parameters, it is crucial to have statistically significant results to support the strong claims in the paper, e.g., "significantly better". A minor concern is that despite that the blended Bellman operator is novel as a whole, the actual algorithmic implementation of each individual operator is not. That being said, I do see the contribution of this work and encourage the authors to provide rigorous empirical results to support the claim in the next revision.

**Justification For Why Not Higher Score:**

The experiments are only with 5 seeds, which is not statistically significant. The empirical implementation of each individual operator is not new.

**Justification For Why Not Lower Score:**

N/A

---

### Decision · Program_Chairs · 2024-01-16

Reject